

# Pre-activation of aeosol particles by pore condensation and freezing

Claudia Marcolli[1,2]

[1]Institute for Atmospheric and Climate Science, ETH, Zurich, Switzerland
[2]Marcolli Chemistry and Physics Consulting GmbH, Zurich, Switzerland

*Correspondence to:* C. Marcolli (claudia.marcolli@env.ethz.ch)

**Abstract.** Pre-activation denotes the capability of particles or materials to nucleate ice at lower relative humidities or higher temperatures compared to their intrinsic ice nucleation efficiency after having experienced an ice nucleation event or low temperature before. This review presumes a pore condensation and freezing (PCF) mechanism to analyze studies on pre-activation. Idealized trajectories of air parcels are used to discuss the pore characteristics needed for ice to persist in pores and

10 to induce macroscopic ice-growth out of the pores. The pore width needed to keep pores filled with water decreases with decreasing relative humidity as described by the inverse Kelvin equation. Thus, narrow pores remain filled with ice well below ice saturation. However, the smaller the pore width, the larger the melting and freezing point depressions within the pores. Therefore, pre-activation by PCF is constrained by the melting of ice in narrow pores and the sublimation of ice from wide pores imposing severe restrictions on the temperature and relative humidity range of pre-activation for cylindrical pores. Ice is

15 better protected in ink-bottle-shaped pores with a narrow opening leading to a large cavity. However, whether pre-activation is efficient also depends on the capability of ice to grow macroscopically, i.e. out of the pore. A strong effect of pre-activation is expected for swelling pores, because at low relative humidity (RH) their openings narrow and protect the ice within them against sublimation. At high relative humidities, they open up and the ice can grow to macrosopical size and form an ice crystal. Similarly, ice protected in pockets are perfectly sheltered against sublimation but needs the dissolution of the surrounding matrix

to be effective. Pores partially filled with condensable material may also show pre-activation. In this case, complete filling occurs at lower RH than for empty pores and freezing shifts to lower temperatures.

Pre-activation experiments confirm that materials susceptible to pre-activation are indeed porous. Pre-activation was observed for clay minerals like illite, kaolinite and montmorillonite with inherent porosity. The largest effect was observed for the swelling clay mineral montmorillonite. Some materials may acquire porosity depending on the formation and processing

conditions. Particles of $CaCO_3$, meteoritic material, and volcanic ash showed pre-activation for some samples or in some studies but not in other ones. Quartz and silver iodide were not susceptible to pre-activation.

Atmospheric relevance of pre-activation by a PCF mechanism may not be generally given but depend on the atmospheric scenario. Lower-level cloud seeding by pre-activated particles released from high-level clouds crucially depends on the ability of pores to retain ice at the relative humidities and temperatures of the air masses they pass through. Porous particles that are

30 recycled in wave clouds may show pre-activation with subsequent ice growth as soon as ice saturation is exceeded after having passed a first cloud event. Volcanic ash particles and meteoritic material likely influence ice cloud formation by pre-activation.



Therefore, pre-activation needs to be considered when ice crystal number densities in clouds exceed the number of ice-nucleating particles measured at the cloud forming temperature.

## 1 Introduction

Ice is the stable phase of water below 273 K, however, micrometer-sized water droplets can be supercooled down to the
homogeneous onset of freezing at about 237 K. There is ample evidence that liquid droplets indeed persist in the atmosphere until they reach low enough temperatures to freeze homogeneously (DeMott et al., 2003; Hoyle et al., 2005; Peter et al., 2006; Krämer et al., 2009). In the presence of ice-nucleating particles (INPs) cloud glaciation can occur at any temperature between ice melting and the onset of homogeneous ice nucleation. Different types of particles are considered important as atmospheric INPs (Hoose and Möhler, 2012). Among these, mineral dust particles are probably best established, but mostly at temperatures
below 255 K (Murray et al., 2012; Atkinson et al., 2013; Kaufmann et al., 2016), which is too low to explain the occurrence of ice and mixed-phase clouds at higher temperatures (Korolev et al., 2003). Biological particles have obtained considerable attention as INPs recently. While they seem to be able to nucleate ice at high temperature, it is uncertain whether they are abundant enough to account for cloud glaciation above 255 K (Hoose et al., 2010; DeMott and Prenni, 2010; Conen et al., 2011; Pummer et al., 2012; O'Sullivan et al., 2014; 2015; Twohy et al., 2016).

Apart from the search for highly active INPs to explain cloud glaciation above 255 K, also the processes leading to ice in clouds need consideration. Different mechanisms of heterogeneous ice nucleation in the atmosphere are discriminated. Probably the most important one is immersion freezing, where a particle nucleates ice from within the body of a supercooled water droplet or solution particle (Murray et al., 2012; Vali, 2015). This mechanism is considered at work when a cloud droplet has formed on an INP earlier on and further cooling is needed until heterogeneous ice nucleation on the particle becomes efficient. This
process is usually discriminated from condensation freezing, which refers to concurrent cloud droplet activation and ice nucleation, although it is not clear whether immersion and condensation freezing can be discriminated from a microphysical point of view. However, in some cases only a condensation freezing process is viable, for example for ice nucleation on soluble INPs. Another interaction of cloud droplets with aerosol particles is collision, either when the particles have remained interstitial during cloud activation or when dry air masses are entrained into a cloud. If a collision between a cloud droplet and an aerosol
particle results in droplet freezing, contact freezing is considered at work. This process has the reputation to occur at higher nucleation rates than immersion freezing for otherwise the same conditions i.e. the same INP at the same temperature, although this does not seem to be the case in general (Ladino Moreno et al., 2013; Nagare et al., 2016). It is not clear whether the supposedly increased ice nucleation efficiency is caused by the collision itself or the position of the particle on the droplet surface. More recent research gives evidence that the position of the particle on the droplet surface is indeed a preferred location
to initiate ice nucleation (Durant and Shaw, 2005; Shaw et al. 2005). However, a particle can also adhere to the droplet surface after cloud droplet activation, if this is the preferred configuration based on the energy balance between surface and interfacial tensions. Nagare et al. (2016) therefore discriminate between collisional contact freezing when the collision is responsible for



ice nucleation and adhesion freezing when the adhesion to the droplet surface enhances the ice nucleation efficiency of the particle.

All the nucleation mechanisms outlined above involve a liquid phase in which ice embryos develop. Yet, there is ample evidence that ice crystals also form below water saturation (e.g. Hoose and Möhler, 2012; Marcolli, 2014). In these cases, ice nucleation

is considered to occur from supersaturated vapor on an ice-nucleating surface without prior formation of liquid (Vali et al., 2015). This view was recently challenged by Marcolli (2014) who theorized that deposition nucleation is in fact pore condensation and freezing (PCF) occurring in voids and cavities that may form between aggregated primary particles or pores that host water below water saturation due to the inverse Kelvin effect. Moreover, ice that persists in pores or cavities below ice saturation should be able to initiate ice crystal growth once relative humidity (RH) exceeds ice saturation.

The capability of particles or materials to nucleate macroscopic ice at lower relative humidites and/or higher temperatures compared to their intrinsic ice nucleation efficiency after having experienced an ice nucleation event or low temperatures before, is termed pre-activation. The first description of pre-activation was for $CdI_2$ particles by Fournier d'Albe (1949). In explanation of this behavior, Mason (1950) suggested that the surface of the $CdI_2$ crystals retained a microscopic oriented film of ice that was kept while the bulk ice sublimated. This ice film could have acted as a nucleating surface when the sample was cooled

again and might have been destroyed when the eutectic temperature of $CdI_2$ solutions (264.6 K) was approached. Mossop (1956) hypothesized that the substances showing pre-activation are singular in that some particles retain a small ice embryo on their surfaces, possibly in suitable cavities. Mason and Maybank (1958) suggested that some materials may hold ice-like aggregates of molecules within the crystal structures or on surface dislocation sites. Higuchi and Fukuta (1966) found that indeed nucleation of macroscopic ice was not needed for pre-activation but exposure to low temperature was sufficient. The fact that pre-activation

is destroyed once the particles are heated above 273 K has often been taken as evidence for the melting of ice embryos rather than the complete sublimation of surface layers (Mossop, 1956; Mason and Maybank, 1958; Roberts and Hallett, 1968). In a theoretical study, Fukuta (1966) therefore concluded that ice forms by homogeneous or heterogeneous nucleation in capillary-held water on almost any insoluble particle with capillaries that are clean and allow water to condense. This ice may then survive exposure to a dry atmosphere. In later studies, the capillary hypothesis was rejected again and the focus turned to ice-like layers

on the surface of INPs. Roberts and Hallett (1968) considered the capillary hypothesis inconsistent with the observation that quite low values of ice supersaturation are necessary to nucleate ice from pre-activated particles. They argued that this would require ice retained at contacts between adjacent particles, which would necessitate large capillaries of unique size, which seemed unlikely to them. Evans (1967) and Edwards et al. (1970) explained pre-activation with the presence of a water layer on the ice-nucleating surface that transforms at a transition temperature to an ordered ice-like state that is supposed to act as an

ice nucleator. Since the transition is subject to hysteresis, the sample has to be supercooled to reach the ordered state. If the ice melting temperature is lower than the transition temperature, the ordered state can be maintained during heating and initiate ice nucleation at a higher temperature during a subsequent cooling cycle and for some materials even above 273 K. This explanation was also suggested by Seeley and Seidler (2001) for pre-activation of droplets that are covered with Langmuir films of aliphatic





alcohols. Edwards and Evans (1971) showed that the same explanation also holds for pre-activation in a gaseous environment when the ice-nucleating material is exposed to relative humidity with respect to ice $RH_i < 100$ % between freezing cycles. Research on pre-activation ceased in the early 1970s and was only resumed three decennials later by Knopf and Koop (2006), who clearly favored the survival of ice in pores and capillaries as the explanation of pre-activation. More recently, Wagner et

al. (2016) analyzed the pre-activation behavior of different particles presuming that pre-activation is due to capillary condensation of supercooled water and subsequent homogeneous freezing.

In this study, we focus on the PCF mechanism as the explanation for pre-activation. Much has been learned about freezing and melting of water in pores, which improved the knowledge of conditions to produce and keep ice in capillaries (Marcolli, 2014). In Sect. 2, we give the theoretical background for nucleation and preservation of ice in pores. Pre-activation of particles and ice

crystal growth along idealized atmospheric trajectories are discussed in Sect. 3. Laboratory studies on pre-activation are critically reviewed and analyzed in terms of pore condensation and freezing in Sects. 4 and 5. Section 6 explores atmospheric situations for which pre-activation might be relevant. A summary and conclusions are given in Sect. 7.

## 2 Theoretical background of nucleation and preservation of ice in capillaries

Recently, melting and freezing of water in confinement gained increasing interest with the availability of new mesoporous

materials and increased capabilities of molecular dynamic simulations (e.g. Moore, 2010; 2012). Frequently used materials for experimental studies are mesoporous silica, zeolites, porous silicon, porous glass, and carbon nanotubes (Alba-Simionesco et al., 2006). Water uptake and release, as well as melting and freezing in pores have been reviewed in Marcolli (2014).

Pores of mesoporous materials fill with water at water sub-saturated conditions in accordance with the inverse Kelvin equation given by:

$$\frac{p_{lc}}{p_l} = \exp(\frac{-4\gamma_{gl}(T)M_w \cos\theta}{\rho_l(T)DkT}). \tag{1}$$

Here, $p_{lc}$ is the water vapor pressure over the concave water surface, $p_l$ is the water vapor pressure over a flat water surface, $\gamma_{gl}(T)$ is the surface tension of water at the air/water interface, $M_w$ the molecular mass of water, $\rho_l(T)$ the density of liquid water, $D$ the diameter of the pore, $\theta$ the contact angle of water on the pore wall (Fukuta, 1966), $k$ is the Boltzmann constant, and $T$ the absolute temperature. Note that in case of perfect wetting ($\theta = 0°$), the pore diameter becomes equal to the diameter of the

25 curved water surface. Using Eq. (1), the onset of capillary condensation in pores of mesoporous silica materials is well described for cylindrical pores. For cage-like pores with small openings and large cavities the diameter of the cavity is predictive for the onset of condensation (Kittaka et al. 2011). Hysteresis between water uptake and release is small for cylindrical pores but much larger for cage-like pores.

The surface tension $\gamma_{gl}(T)$ and the density of liquid water $v_l(T)$ are both temperature dependent. The temperature dependence of

30 the surface tension can be described with the IAPWS (International Association for the Properties of Water and Steam) correlation (Hrubý et al., 2014; Vinš et al., 2015):



$$g_{gl}(T) = B\tau^{\mu}(1 + b\tau) \tag{2}$$

with $\tau = 1 - T/T_c$ being the dimensionless distance from the critical temperature $T_c = 647.096$ K, $\mu = 1.256$ being a universal critical exponent, and coefficients $B$ and $b$ having values of 235.8 mN·m$^{-1}$ and -0.625, respectively.

Recent measurements of the density of supercooled water in pores by X-ray diffraction yielded values in the range of 0.9 – 1.01 gcm$^{-3}$ for $T = 100 – 300$ K with a minimum around 200 K and a maximum at approximately 277 K (Liu et al., 2013; 2015). Based on this data and bulk measurements (Hare and Sorensen, 1987; CRC Handbook of Chemistry and Physics, 2015) the following parameterization was derived for the density of supercooled water in gcm$^{-3}$ and with a validity range from 50 to 393 K:

$$\rho_l(T) = 1.8643535 - 0.0725821489 \cdot T + 2.5194368 \cdot 10^{-3} \cdot T^2$$
$$-4.9000203 \cdot 10^{-5} \cdot T^3 + 5.860253 \cdot 10^{-7} \cdot T^4$$
$$-4.5055151 \cdot 10^{-9} \cdot T^5 + 2.2616353 \cdot 10^{-11} \cdot T^6 \tag{3}$$
$$-7.3484974 \cdot 10^{-14} \cdot T^7 + 1.4862784 \cdot 10^{-16} \cdot T^8$$
$$-1.6984748 \cdot 10^{-19} \cdot T^9 + 8.3699379 \cdot 10^{-23} \cdot T^{10}.$$

When water melts or freezes in pores, the melting and the freezing temperatures are depressed compared to the values measured in bulk water. The pore diameter needed to preserve ice in confinement at temperature $T$ can be related to the critical radius $r_c(T)$ for which the growth of an embryo becomes equal to the probability of decay (Vali et al., 2015):

$$r_c(T) = \frac{2\gamma_{sl}(T)v_s(T)}{kTln\frac{p_l}{p_s}} \tag{4}$$

In this expression, $p_s$ is the vapor pressure over ice, $\gamma_{sl}(T)$ is the interfacial tension between ice and water and $v_s(T)$ is the volume of a H$_2$O molecule in ice. Parameterizations for $\gamma_{sl}(T)$ and $v_s(T)$ are given in Zobrist et al. (2007). To incorporate a cluster of critical radius, a pore needs a diameter $D_p = 2r_c + 2t$ with $t$ being the width of a quasi-liquid layer between pore wall and ice embryo with a typical value of 0.6 nm (Marcolli, 2014). If the pore diameter is less than $D_p$ at a given temperature, no pore ice can form, because the pore is too narrow to enable the ice embryo to grow to critical size.

## 3 Idealized scenarios of pre-activation

In Figures 1 – 3 pre-activation scenarios along idealized atmospheric trajectories are displayed for different pore types. Starting conditions are T = 273 K and RH$_w$ = 30 % for wet trajectories and T = 273 K and RH$_w$ = 1 % for dry trajectories. Air parcels are assumed to rise (dry) adiabatically until condensation sets in and a cloud is formed. In the case of wet trajectories, a liquid cloud forms at 100 % RH$_w$. Further cooling leads to heterogeneous ice nucleation on a nucleation site located on the particle surface that becomes active at 255 K. Ice crystal growth is assumed to decrease RH$_i$ to 100 % at constant temperature. Warming due to latent heat release is neglected. In the case of dry trajectories, a cirrus cloud is supposed to form by PCF and the relative humidity is reduced to 100 % RH$_i$. For both wet and dry trajectories, ice may remain in pores at RH$_i$ < 100 % and initiate ice





crystal growth when relative humidity increases again above ice saturation. The following discussion of pre-activation scenarios assumes thermodynamic equilibrium between vapor, ice and liquid water phases. Transient persistence of evaporating pore ice is neglected. Hence, the time an aerosol particle spends along a trajectory is not relevant. The investigated pore types are sketched in Fig. 4. All given pore sizes are in diameter if not stated otherwise.

Figure 1 outlines the phase changes of an aerosol particle that contains a cylindrical pore of 8 nm diameter and acts as INP in immersion mode at 255 K. At the start of the wet trajectory, the pore is empty and only fills at $T = 261.2$ K and $RH_i = 83$ %. When water saturation is reached at 257.5 K the particle activates to a cloud droplet. Cooling of the air parcel causes the particle to freeze at 255 K in immersion mode, allowing it to grow into an ice crystal while $RH_i$ in the cloud is reduced to 100 %. Adiabatic heating is assumed to occur when the air parcel sinks and leads to the sublimation of ice crystals when $RH_i$ falls below

100 %. Pre-activation is lost at $T = 256.2$ K when ice within the 8 nm pore melts because of the melting point depression in confinement. The liquid water within the pore sublimates at 256.5 K. For this scenario, pre-activation is restricted to only slight drying to $RH_i \cong 90$ %.

In the case of the dry trajectory, ice saturation is reached at 226.7 K when the pore is still empty. Water condenses in the pore at $RH_i = 106$ % ($T = 226.2$ K) and immediately freezes by homogeneous nucleation. When $RH_i$ is decreased below 100 % due

to adiabatic heating, ice in the pore sublimates together with the bulk ice. Therefore, in a cylindrical pore of 8 nm, no pre-activation occurs for $T < 233$ K because of the sublimation of the pore ice.

Figure 2 outlines pre-activation scenarios of a particle with an ink-bottle-shaped pore with a pore opening of 4 nm in width and 2 nm in length which leads to a cavity of 20 nm (see Fig. 4). The particle is assumed to contain a nucleation site on its surface that is active in immersion mode at $T \leq 255$ K. Pores with large cavities and narrow openings may occur in porous particles or

in aggregated particles as inter-particular voids (Roberts and Hallett, 1968). Indeed a large fraction of airborne and surface collected dust particles seem to be present as aggregates of different minerals (Reid et al., 2003). While ink-bottle shaped pores might acquire a liquid plug at the pore opening already at low RH, water adsorption isotherms of the mesoporous silica material SBA-16 with cage-like pores, have shown that water adsorption depends on the diameter of the cavity (Kittaka et al., 2011). Pore filling along the idealized trajectories is therefore assumed to occur according to the inverse Kelvin equation applied to the

pore cavity diameter instead of the pore opening. Comparison with the data of Kittaka et al. (2011) indicates that using the inverse Kelvin equation along with the cavity diameter results in a slight overestimation of $RH_w$ needed for pore filling (Marcolli, 2014). Along the wet trajectory, the cavity with a diameter of 20 nm fills with water at 259 K when $RH_i$ just passes 100 %, as shown in Fig. 2. Water saturation is reached at $T = 257.5$ K and the particle is supposed to activate as a cloud droplet. Further cooling leads to ice nucleation in the immersion mode at $T = 255$ K, followed by ice crystal growth, which reduces $RH_i$

to 100 %. Freezing experiments with SBA-16 (Kittaka et al., 2011) showed that ice does not propagate through cage connections with diameters of 3.9 nm when $T > 245$ K. Hence, the water confined in the pore most probably remains liquid because the bulk ice at the surface of the particle cannot propagate through the narrow pore opening of 4 nm at $T = 255$ K. Therefore, no ice is formed in the pore and the particle does not become pre-activated.





Along the dry trajectory, water vapor condenses in the 20 nm cavity at $RH_i = 135$ % and T = 224 K, immediately followed by homogeneous ice nucleation within the pore leading to the growth of an ice crystal. With a width of 4 nm and a length of 2 nm, it is assumed that the pore opening is not able to host ice on its own. The ice crystal sublimates when $RH_i$ sinks below 100 %. For the scenario of adiabatic heating of the air parcel, ice remains within the cavity protected by the water in the narrow pore

opening of 4 nm as long as $RH_i > 70$ %. If the air parcel is cooled adiabatically again before $RH_i$ falls below 70 %, the ice in the pore is expected to initiate ice crystal growth when $RH_i > 100$ % since at this low temperature, ice propagation through the pore opening should readily occur. If we assume a trajectory of the particle as sketched by the red dashed line, ice should survive in the pores when ice saturation is reached at 238 K. If the relative humidity rises again, the ice contained within the pore can initiate macroscopic ice crystal growth for $RH_i > 100$ % because at this low temperature, ice should be able to propagate through

the pore opening. However, a cylindrical pore with 4 nm diameter should have the same ability of pre-activation as a cylindrical pore of the same width. Therefore, an ink-bottle-shaped pore is probably not better suited for pre-activation at low temperatures than a cylindrical pore with the width of the pore opening of the swelling pore.

Figure 3 shows pre-activation scenarios for a swelling ink-bottle-shaped pore with a cavity of 20 nm and a pore opening width that depends on relative humidity and particle history. The pore is supposed to swell when it fills with water. Such pore swelling

has been described for the inter-particular voids that form between aggregated particles of e.g. montmorillonite (Salles et al., 2008). For simplicity, it is assumed that pore swelling widens up the pore opening and has no effect on the cavity diameter, which is kept constant at its initial value of 20 nm (see Fig. 4 for pore geometry). It is further assumed that sublimation of pore water or pore ice leads to a contraction of the pore to such a degree that capillary forces acting on the pore opening keep the pore filled. The minimum pore opening is assumed to be 1 nm in diameter and is realized when the pore is empty. Again, the

particle is supposed to act as INP in immersion mode at 255 K. Along the wet trajectory the pore fills with water at $RH_i \cong 101$ % and $T \cong 259$ K leading to a widening of the pore opening. Water saturation is reached at 257.5 K, a liquid droplet forms and the pore is supposed to widen even more. At 255 K nucleation in immersion mode leads to the freezing of the whole droplet including pore water and subsequent ice crystal growth. When relative humidity falls below 100 % $RH_i$, the ice crystal evaporates leaving behind the particle that still contains ice in the pore. The pore ice is supposed to sublimate partly such that

the pore opening shrinks to such a degree that capillary forces remain strong enough to keep the pore opening filled with a liquid plug. The pore ice is therefore protected against sublimation, however, it will melt if the temperature increases above 267 K because of the melting point depression in confinement. When the air parcel is cooled again adiabatically before this temperature is reached, the ice confined in the pore can initiate ice crystal growth along the adiabatic trajectory when $RH_i > 100$ %, assuming that the pore opening widens up in response to the RH increase. Otherwise, relative humidity has to reach water saturation and

ice crystal growth will set in when the pore ice is released upon CCN activation. A strong effect of pre-activation is achieved when relative humidity is increased at a constant temperature of 266 K, which is just below the temperature for pore ice melting. In this scenario, outlined by the green dashed line, ice crystal growth starts just when relative humidity exceeds ice saturation.



Along the dry trajectory, the pore fills with water at $RH_i \cong 135$ % and $T \cong 224$ K leading to an increase of the pore opening and immediate freezing of the pore water. For pore openings $\geq 3.5$ nm, pore ice can propagate through the pore opening and initiate ice crystal growth (Marcolli, 2014). When the air parcel is adiabatically warmed again, bulk ice sublimates at $RH_i < 100$ %. The pore ice is supposed to remain protected by the liquid water plug in the pore opening while the pore opening shrinks to its

minimum size of 1 nm. For $RH_i < 5$ % even water in 1 nm pores evaporates, which is the case at $T \cong 247$ K for the dry trajectory. When RH is increased again before such dry conditions are reached and assuming that the pore opening widens up with increasing relative humidity, the pore ice should be able to initiate growth of an ice crystal as soon as ice saturation exceeds 100 %. Both, wet and dry trajectories illustrate that swelling pores that presumably arise in aggregated particles with inter-particular voids, remain pre-activated up to high temperatures and resist low relative humidity.

The pre-activation scenarios outlined above apply to empty pores. Conditions of pore filling and emptying as well as freezing and melting of pore ice are modified in the presence of coatings. Pores and cracks are supposed to fill with condensable material due to the inverse Kelvin effect even before the particle acquires a coating (e.g. Sjogren et al., 2007). Once a particle has obtained a complete coating, pore condensation and freezing becomes insignificant because then the entire coating responds to humidity changes with continuous water uptake and release depending on its hygroscopicity. Before a complete coating is

reached, partly filled pores take up water gradually. The relative humidity of complete pore filling and emptying is lowered compared with the case of pores containing no condensed soluble material because soluble material in the pores lowers the water activity of condensing water. On the other hand, freezing and melting in pores filled with an aqueous solution occurs at a lower temperature compared to the pure water case (e.g. Sjogren et al., 2007). When the cylindrical pore of the particle shown in Fig. 1 contained enough condensable material to form an aqueous solution with a water activity of 0.95 when it is full, filling

occurs at $T = 226.7$ K and $RH_i = 100$ % along the dry trajectory instead of filling of the empty pore at 106 % $RH_i$. A solution with $a_w = 0.95$ causes a freezing point depression to 226.2 K (Koop et al., 2000). When this temperature is reached, the pore water finally freezes and the solute forms a freeze concentrated solution presumably at the walls and in the opening of the pore (see Fig. 4). The pore ice initiates ice crystal growth so that $RH_i$ decreases to 100 %. When the particle is adiabatically warmed again along the trajectory, the solution plug at the pore opening remains in equilibrium with the water vapor in the air and further

concentrates. A freeze concentrated solution at $226 - 229$ K in equilibrium with ice has $a_w = 0.64 - 0.66$. When $RH_i$ decreases below 66 % the ice starts to melt, the water evaporates, and pre-activation is lost. This scenario shows that pre-activated pores containing some condensable material might resist emptying to drier conditions than pores without condensable material in them. However, the exact effect depends on the degree of filling and the hygroscopicity of the condensing material.

For the wet trajectories, a cloud droplet forms before freezing occurs. Because dilution of the coating in the cloud droplet is

large, the dissolved coating material does not cause a freezing point depression. When the droplet freezes, the dissolved material is expelled from the ice and gathers on the surface. When the ice sublimates upon warming, the dissolved material distributes most probably on the particle surface. Unless it forms plugs on ice containing pores, it is not expected to influence pre-activation.





If it collects on the pore opening, it can influence pre-activation in different ways. It might protect ice against sublimation, however, it might also hinder ice growth out of the pore.

Finally, pre-activation can occur when particles turn glassy or effloresce and water trapped in pockets during this process freezes upon further cooling. Ice in completely enclosed pores is preserved also under dry conditions. Pre-activation is lost when

particles become liquid by glass transition or deliquescence. In the case of frozen solution pockets, ice crystal growth can start when the particle dissolves and the trapped ice is released. For crystalline particles, Wagner et al. (2014) referred to this process as deliquescence-induced ice growth. When the pores have connections to the surface, ice crystal growth may be initiated by the pore ice as soon as the air becomes supersaturated with respect to ice. Wagner et al. (2014) called this process depositional ice growth. A detailed description of these mechanisms can be found in Wagner et al. (2012, 2014). Formation of highly porous

aerosol particles by atmospheric freeze-drying in ice clouds has also been discussed by Adler et al. (2013). Their Figure 1C shows a cross-section through a freeze-dried glassy NOM (natural organic matter) particle revealing embedded pores with diameters of up to 200 nm.

## 4 Laboratory studies on pre-activation

In the following, laboratory studies on pre-activation are reviewed under the presumption that pre-activation occurred by the PCF mechanism.

### 4.1 Expansion chamber experiments by Fournier d'Albe (1949)

Fournier d'Albe (1949) was the first to describe pre-activation. He investigated cadmium iodide ($CdI_2$), sodium chloride (NaCl), sodium nitrate ($NaNO_3$), and cesium iodide (CsI) in an expansion chamber with a volume of 2 l. Aerosol particles with a number

density of 500 – 1000 $cm^{-3}$ and diameters between 0.1 and 1 µm were produced from sprayed dilute solutions. Expansions were performed close to adiabatic conditions. Liquid droplets and ice crystals were detected in the light of a mercury arc lamp with a detection limit of 1 $cm^{-3}$. During a first expansion, the particles acted as CCN and produced a liquid cloud at T > 232 K. A fog consisting almost completely of ice crystals formed when final expansion temperatures fell below 232 K and water saturation was reached. Recompression led to the sublimation of the ice crystals. Pre-activation was observed during a second expansion

in the case of cadmium iodide, a deliquescing salt, but not for the other investigated salts. If temperature was kept < 264 K during recompression, ice crystals grew again on $CdI_2$ particles during a second expansion well below water saturation when temperature fell about 1 K below the frost point (see also Table 1). Fournier d'Albe (1949) reports the freezing point of a saturated solution of $CdI_2$ at 264.65 K, which corresponds to the eutectic melting temperature. Apelblat and Korin (2007) measured water activities of saturated $CdI_2$ solutions of $a_w = 0.902$ and 0.947 for 281.05 K and 298.15 K, respectively. Assuming

a deliquescence $RH_w < 90$ % for T < 273 K, the particles are supposed to deliquesce at $RH_i < 100$ % i.e. before ice saturation is reached when T > 262 K. At lower temperatures, the particles should still be solid at ice saturation. This might explain why pre-activation was lost when particles were heated above 264 K. Pores require diameters of 14 nm at T = 264 K to preserve ice. For



narrower pores, ice is lost due to the melting point depression in confinement. Assuming cylindrical pore shapes, pores remain filled when relative humidity with respect to ice is kept larger than 91 % at 264 K. Since the chamber walls were covered with ice, relative humidity likely stayed close to ice saturation. The presence of pores with diameters ≥ 14 nm in CdI$_2$ particles, could therefore explain the pre-activation observed by Fournier d'Albe. As an alternative explanation for the higher freezing

temperatures during the second expansion, Edwards et al. (1970) proposed the crystallization of a CdI$_2$ hydrate, which they claimed to be an excellent ice nucleator. Fournier d'Albe investigated whether cesium iodide, another soluble salt with $a_w$ = 0.947 at 278 K for a saturated solution (Apelblat and Korin, 2006) would show a similar ability for pre-activation. However, during a second expansion no ice crystals could be observed when the temperature remained > 232 K. In addition, Fournier d'Albe investigated silver iodide particles produced by heating the salt on a platinum wire. The aerosol activated to a mixed fog

for T < 262 K and a fully glaciated fog formed at T < 256 K. However, there was no pre-activation observed. Also experiments with outdoor air in the expansion chamber did not reveal pre-activation.

## 4.2 Expansion chamber experiments by Mossop (1956)

Mossop (1956) used the same setup as Fournier d'Albe (1949) but with higher particle number concentrations. Out of 50 non-specified materials, only CdI$_2$, CaCO$_3$ (Iceland spar), gypsum (CaSO$_4$x2H$_2$O), and Na bentonite clay showed pre-activation (see

Table 2). He carried out experiments with CdI$_2$ particles that he produced either from spraying of aqueous solutions or by heating a small amount of the salt on a platinum wire to give a visible smoke. Pre-activation occurred up to 264 K for sprayed dilute solutions, corroborating the experiments by Fournier d'Albe (1949). Particles produced from smoke lost their pre-activation ability already at 261 K, which he explained by the smaller size of particles produced this way. Pre-activation was observed for atomized aqueous suspensions of Iceland spar (calcite) of analytical purity (<0.1 % impurity) but not for a CaCO$_3$

sample presumably produced by precipitation (Analar). Pre-activation was maintained for heating to 269.5 K between the first and second expansions for the Iceland spar. Gypsum and Na bentonite clay – a clay mineral with montomorillonite as the main component and minor shares of quartz, mica, feldspar, pyrite and calcite – also showed pre-activation for warming up to 267.8 K and 266.6 K, respectively. To retain ice in pores up to such high temperatures, pore diameters > 20 nm are required.

## 4.3 Freezing chamber experiments by Day (1958)

Day (1958) tested the ability of different materials to show pre-activation in a freezing chamber maintained at 264 K. Particles were injected with a gun, so that an adiabatic expansion from bursting pressure to atmospheric pressure cooled the air within the gun rapidly by at least 70 K leading to immediate freezing of condensed water drops. To sublimate the ice, particles were kept for some minutes at 264 K at 84 – 98 % RH$_i$ followed by humidification to investigate pre-activation. Results of ball milled Iceland spar in the size range from 1 – 15 µm with large numbers from 1 – 3 µm were presented in most detail: 1 – 5 % of the

particles showed pre-activation when kept for 1 min at 84 – 98 % RH$_i$ (see Table 3). However, the RH when ice crystals started to grow from the pre-activated particles is not clearly stated. Pre-activation became negligible when the time under sub-saturated conditions was extended to 5 min or the humidity was lowered by drying with a saturated CaCl$_2$ solution (the deliquescence



relative humidity of $CaCl_2$ hexahydrate is 32 % RH at 293 K). Meteorites (4 different aerolites) showed a pre-activated fraction up to 0.005, which decreased to ~$10^{-5}$ within a minute. No or negligible pre-activation was found for AgI, volcanic rock, quartz, mica, clay and gypsum. At 264 K, pores > 14 nm are needed to prevent ice from melting in confinement. Cylindrical pores of this size are not supposed to retain water at $RH_i$ < 92 %. This confirms the experimental findings that pre-activation was a

transient effect and not sustained under equilibrium conditions.

## 4.4 Cloud chamber experiments by Mason and Maybank (1958)

Mason and Maybank (1958) tested 28 naturally-occurring mineral dusts for their ability to nucleate ice. The minerals were ground with a pestle and mortar leading to an appreciable fraction of micron and sub-micron particles. The experimental setup consisted of two thermostatically controlled chambers. Supercooled clouds were formed by evaporation from a piece of water-

soaked gauze, heated by a small electric bulb. Crystal growth could be observed directly in the chamber and roughly quantified with films on which the crystals deposited with a detection range of 1 – 100 per liter of air. The materials were introduced as a fine dust into the first chamber through a side tube. Ice crystals, activated in this first chamber were drawn through a 50 cm long glass tube into a second chamber within 10 – 30 s. The transit tube was surrounded by an ice and salt bath and held at temperatures between 268 K and 274 K and a relative humidity of about 32 % RH given by the presence of calcium chloride as

the drying agent. If no pre-activation occurred under these conditions, the dust was considered not susceptible to pre-activation. Ten out of the 28 investigated mineral dusts showed pre-activation, the rest did not respond (see Table 4). Among the responding ones were clay minerals (kaolinite and montmorillonite), volcanic ash and one out of three stony meteorites, which all had similar compositions, namely ~90 % silicates (mainly pyroxene) and the remainder chiefly iron oxide. Variation of the temperature of the conditioning tube between 268 and 274 K had little effect, but for $T_{cond}$ = 274 K pre-activation was lost. The

drying conditions in the connecting tube seem to have lasted long enough for bulk ice to sublimate and short enough to keep ice present in pores. Pores with diameters of 26 nm, 36 nm, 50 nm, 100 nm, and 200 nm exhibit melting point depressions of about 4 K, 3 K, 2 K, 1 K and 0.5 K, respectively. To retain water in such large pores, $RH_w$ would have to remain above 90 %. From this, it is clear that total sublimation of pore ice was prevented by keeping the conditioning time in the transition tube too short to reach thermodynamic equilibrium.

## 4.5 Cold stage experiments with freshly-cleaved fluor-phlogopite mica by Layton and Harris (1963)

Layton and Harris (1963) observed pre-activation on freshly-cleaved, synthetic fluor-phlogopite mica mounted on a microscope in a cold chamber with independent temperature and humidity control. Relative humidity was regulated by a temperature controlled ice surface on the bottom of the chamber. The rate of temperature change was one or two degrees per minute. During the first cooling, growth of ice only occurred at water saturation and was preceded visibly by liquid droplet formation for T >

247 K. After the ice was sublimated from the mica surface, re-cooling led to ice growth at steps on the crystal at very little supersaturation with respect to ice. When the sample temperature was raised above freezing, the growth would occur at water saturation generally over the whole crystal surface. Best re-nucleation sites could be identified in connection with small pieces



of mica which remained stuck on the mica surface during cleavage and possibly led to deep and sharp recesses in which ice could survive.

## 4.6 Cold stage experiments with deposited particles by Higuchi and Fukuta (1966)

Higuchi and Fukuta (1966) were able to pre-activate particles without forming macroscopic ice on them. They deposited the particles onto a metal foil in a small air-tight plastic box and cooled them to conditioning temperatures between 238 K and 203 K in the presence of silica gel as the drying agent to keep $RH_w$ at about 40 %. After cooling, the samples were warmed up and the ice forming ability of the particles was detected by supplying water vapor. They investigated different clay minerals (1 – 10 µm fraction), silica gel, stony meteorite, volcanic ash, clay and soil. All tested samples showed pre-activation at T = 270 – 271 K. Pre-activation was retained for more than two months when the samples were kept at T < 273 K and a humidity below ice saturation but lost after warming above 273 K. For montmorillonite, volcanic ash from Mt. Agung (Bali) and kaolinite, the fraction of active particles was quantified for different pre-conditioning and crystallization temperatures (see Table 5 for details). For a conditioning temperature of 203 K only 0.1 % of the montmorillonite and volcanic ash particles were active at 270 – 271 K compared to more than half at 258 K. This tendency was even more pronounced for kaolinite: nearly 50 % of the particles formed ice at 253 K, but only 10 % at 258 K. For montmorillonite and volcanic ash, pre-activated fractions at 258 K were only about 0.01 for a conditioning temperature of 253 K and increased to 0.15 – 0.75 for conditioning below 233 K. Again, pre-activation was retained for several hours. Because particles remained pre-activated for such long time, it is reasonable to assume that ice in the pores was thermodynamically stable. Higuchi and Fukuta (1966) assumed $RH_w$ = 40 % controlled by silica gel as the conditioning RH. Although there are inconsistencies in their description of the RH control, we keep to this value for the further argumentation. To keep pores filled at $RH_w$ = 40 % and T = 203 K, pore diameters < 4 nm are needed. However, ice in such narrow pores melts at T < 239 K. To avoid pore ice melting at 253 K, 258 K, 270 K, and 271 K pore diameters larger than 7 nm, 9 nm, 39 nm, and 56 nm are required. Such wide pores should be empty at 40 % $RH_w$ and thus cylindrical pores are not suited to show pre-activation at these high temperatures. Ink-bottle-shaped pores with pore openings < 4 nm diameter would be able to keep ice in the cavity but cannot induce macroscopic ice growth at the developing temperatures because the ice cannot propagate out of the cavity. Therefore, pores that swell at high RH with narrow openings and large cavities are needed. Such pores can form as voids between aggregated particles with openings that widen at high RH so that ice can propagate through them even before water saturation is reached. When water droplets form at water saturation, aggregates may also break up, releasing the ice contained between them and thus initiate freezing.

The experiments by Higuchi and Fukuta (1966) support pore condensation and freezing as the reason for pre-activation. Firstly, the share of pre-activated particles decreases with increasing temperature in accordance with the scarcity of particles with wide enough pores to retain ice at high temperatures; secondly, there is a strong increase of the pre-activated fraction when the conditioning temperature decreases below the homogeneous freezing threshold, thirdly, the pre-activated fractions remain almost constant below the homogeneous freezing temperature threshold, and lastly, pre-activation is completely lost above 273 K. All these findings are in accordance with pore ice as the reason for pre-activation. However, as the particles were deposited





on glass cover slips, the locations of pore ice were probably voids between the substrate and the particles rather than pores within the particles or between particle aggregates (see also Sect. 5.1).

### 4.7 Cold stage experiments with deposited particles by Roberts and Hallett (1968)

Roberts and Hallett (1968) observed pre-activation for kaolinite, montmorillonite, Wyoming bentonite, surface glacier debris, stony meteorite, gypsum, calcite, vaterite, and albite. In their experiments, they deposited particles, on a microscope cold stage with independent control of temperature and relative humidity. Substances were ground with mortar and pestle to a mean size of $1 - 2$ µm. About $10^4$ particles, typically aggregates with diameters of $0.5 - 3$ µm could be viewed on one cover slip, so that the threshold of observable ice nucleation activity was taken as the appearance of one ice crystal in $10^4$ particles. Pre-activation investigations were carried out by cooling the sample until ice crystals formed on all particles present. The temperature of the sample was then raised until all the macroscopic ice had completely sublimated. Temperature and humidity were maintained at a constant value for long periods of time up to one week. The particles were tested for ice crystal growth by slowly increasing RH until ice crystals or water drops appeared on the cover slip. Conditions of initial ice crystal formation and after pre-activation for activated fractions of $10^{-4}$ and $10^{-2}$ at water saturation, and threshold nucleation below water saturation are given in Table 6. Montmorillonite and Wyoming bentonite were very efficient after pre-activation and developed ice at 268 K after having been dried at 20 % $RH_i$. To be filled with water at $RH_i = 20$ % at $T = 260 - 270$ K pores need to be very narrow with diameters smaller than 1.5 nm. Such pores are too narrow to freeze or to retain ice at $T > 210$ K. To prevent ice melting at $T = 268$ K pores need diameters of at least 24 nm. Therefore, swelling ink-bottle shaped pores are needed. Again, the particles were deposited on glass cover slips. Therefore, it is likely that water gathered between the substrate and the particles enhanced the intrinsic ice-nucleation ability of the particles (see Sect.5.1). Silver iodide was among the view samples that failed to show pre-activation.

### 4.8 Cold stage experiments with deposited particles by Edwards and Evans (1971)

Edwards and Evans (1971) performed pre-activation experiments with a humidity controlled cold stage. The samples were ground in a mortar, then dispersed in air and allowed to settle on cover slips rendered hydrophobic. In a first step, the crystallization temperature for the activation of 0.1 % of the particles at $RH_w = 120$ % was determined by lowering the temperature to the target temperature at dry conditions and then admitting humidity. For pre-activation experiments, the sample was cooled under dry conditions to a target temperature of typically 243 K and $RH_w$ was raised to over 100 %. The conditioning occurred at the recrystallization temperature by drying at 80 % or 40 % $RH_w$ for 2 min or 45 min, followed by the recrystallization step which was carried out by raising $RH_w$ to 120 %. The number of ice crystals which formed within 5 min was counted. Results are shown in Table 7. $HgI_2$, gypsum, muscovite, silica gel, calcite, kaolinite, alumina, phloroglucinol dihydrate, α-phenazine, l-asparagine, egg albumin, and benzil all showed pre-activation; for $PbI_2$, AgI and CuI, no effect was observable. For $HgI_2$, silica gel, and phloroglucinol dihydrate, dry activation at 80 % $RH_w$ was also tested, which showed no difference compared with wet activation, corroborating the findings by Higuchi and Fukuta (1966). Edwards and Evans (1971)



found pre-activation for most investigated samples, but similarly to Higuchi and Fukuta (1966) and Roberts and Hallett (1968), the particles were deposited on cover slips.

### 4.9 Cold stage experiments with deposited ATD by Knopf and Koop (2006)

Knopf and Koop (2006) monitored in a humidity controlled cold stage individual ice nucleation events on Arizona test dust (ATD) particles with diameters between $0.1 - 10 \, \mu m$ on a hydrophobic substrate in the temperature range of $200 - 260$ K. They exposed the particles to increasing relative humidity at constant temperature and noted the $RH_i$ when ice crystals started to grow. To investigate pre-activation, they reduced $RH_i$ to $5 - 40$ % after having observed ice formation during a first nucleation event and subsequently increased it until ice crystals formed again. In most experiments, the particles which nucleated ice first in the initial experiment, nucleated ice also first in the second one, but at a $0 - 30$ % lower $RH_i$. (see Table 8). Pre-activation ceased when $RH_i$ was reduced to $0.2 - 3.5$ % after the initial nucleation event. They also observed pre-activation for $H_2SO_4$ coated ATD particles. To keep water in pores at 40 % $RH_i$ and 5 % $RH_i$, pores have to be smaller than 2.5 nm and 0.8 nm, respectively. At temperatures $< 220$ K, ice might remain stable in pores with diameters of about 2.5 nm and initiate ice crystal growth. At higher temperatures or for even narrower pores, ice melts because of the melting point depression in confinement. Ice in swelling pores or trapped in spaces between the particle and the substrate are therefore needed to enable ice crystal growth out of the pores.

### 4.10 AIDA chamber experiments by Wagner et al. (2012)

Wagner et al. (2012) investigated pre-activation of aerosols consisting of raffinose, 4-hydroxy-3-methoxy-DL-mandelic acid (HMMA), levoglucosan, a multi-component mixture of raffinose with five dicarboxylic acids (malonic acid, DL-malic acid, maleic acid, glutaric acid, and methylsuccinic acid) and ammonium sulfate (raffinose/M5AS) in the aerosol and cloud chamber AIDA (Aerosol Interaction and Dynamics in the Atmosphere). Prior to an experiment, the inner walls of this 84.3 m$^3$ stainless steel vessel were coated by a thin ice layer. Aerosol particles were generated from injection of dilute aqueous solutions of the investigated compounds yielding particles in the size range from 0.02 to 2 µm diameter. Expansion cooling by controlled pumping led to an increase of relative humidity mimicking rising air parcels in the atmosphere. During a first expansion run, raffinose and HMMA particles nucleated ice homogeneously at $RH_i > 138$ % and $T \leq 230$ K. The relative humidity at which ice crystals appeared was significantly lower when the aerosol was preprocessed in a preceding expansion run, such that the particles froze homogeneously and the freeze concentrated solution vitrified when the temperature fell below the glass transition temperature of the freeze concentrated solution, i.e. Tg' (see Table 9). Typically, $10 - 35$ % of the particles that nucleated ice homogeneously in the first expansion run induced ice nucleation at $105 - 112$ % $RH_i$ in the subsequent expansion run due to pre-activation. In-between expansion runs, they experienced $RH_i = 70 - 80$ %. When the particles did not pass Tg' during homogeneous freezing in the first expansion run, the freeze concentrated solution in the particles remained liquid and the ice nucleation ability was unchanged during the second expansion run. This was the case for the raffinose/M5AS mixture when homogeneous freezing occurred at ~219 K. When the raffinose/M5AS particles froze at ~211 K, the freeze concentrated solution





is supposed to become highly viscous and some particles were therefore susceptible to pre-activation. Pre-activation disappeared when the chamber temperature was raised above the glass transition temperature of the substance under investigation, but remained even if the aerosol was kept for 2.5 h below ice saturation with a minimum $RH_i$ value of 70 %. Wagner et al. (2012) hypothesized that vitrification in the presence of ice crystals may leave behind a structured surface with defects or pores filled

with ice. Cylindrical pores with diameters < 4 nm or ink-bottle-shaped pores with pore openings < 4 nm should indeed remain filled at $RH_i$ = 70 % and preserve ice at T = 210 – 230 K. When the aerosols are composed of water soluble organic substances, the pores rather fill up with an aqueous solution instead of pure water, inducing a decrease of water vapor pressure such that also larger pores remain filled with a solution under the dry conditions between expansion runs. Evidence for the role of pores in the pre-activation of glassy aerosol particles is given by the fact that homogeneous nucleation has to take place close to glass

transition to pre-activate the particles. Crystal growth in highly viscous media leads to dendritic ice crystals (Gránásy et al., 2004; Ciobanu et al., 2010; Song et al., 2012; Giri et al., 2013). Dentritic ice crystals may imprint their structure on the freeze concentrated solution when the particles turn glassy. The glassy particles will keep this structure when the ice sublimates between expansion runs. Particles should therefore become porous when they vitrify around branched ice crystals that subsequently sublimate.

## 4.11 AIDA chamber experiments by Wagner et al. (2014)

Wagner et al. (2014) investigated pre-activation in the AIDA cloud chamber with crystallized ammonium sulfate, oxalic acid, and succinic acid particles (diameters of approximately 0.1 – 2 µm) in the temperature range from 244 to 267 K. As reference experiments, liquid clouds were formed during expansion runs at temperatures where the effloresced particles proved to be not active as INPs in previous studies (Zobrist et al., 2006; Wagner et al., 2010; 2011). Ice crystal growth

could be triggered by temporarily cooling the crystallized particles to a lower temperature before performing the expansion run (see Table 10). Wagner et al. (2014) ascribed the pre-activation of the particles to pockets and pores of aqueous solution within the crystalline material that formed when the solution droplets effloresced after injection into the dry chamber. During the cooling of the chamber, the water in the solution pockets and pores froze. In the subsequent expansion runs, this ice initiated depositional ice growth, when ice in pores that connect to the surface was involved, and freezing during deliquescence, when

completely shielded ice in pockets was freed. Ice crystal growth on pre-activated particles via depositional ice growth occurred with ice active fractions from 0.01 to 0.04 and via deliquescence-induced ice growth with ice active fractions from 0.04 to 0.2. Pre-activation disappeared above the eutectic temperature, which for the organic acids is close to the melting point of ice. The succinic acid aerosol was brought to 227.3 K at $RH_i \geq$ 72 % to freeze the pore water. To keep cylindrical pores filled with pure water at 72 % $RH_i$, the pore diameter should be < 4.1 nm. However, to keep pore water frozen up to ~246 K, diameters > 5 nm

are needed. If pores are of slightly conical shape and/or the lowest RH in the chamber is not maintained for a long time, then PCF is the likely reason for pre-activation. Similarly, oxalic acid was brought to 244 K at $RH_i \geq$ 73 % to freeze the pore water by heterogeneous nucleation on oxalic acid particles acting as INPs (Wagner et al., 2011). Because of the low solubility, freezing and melting point depressions are not important for oxalic acid. Only pores with diameters < 4.7 nm fill with water at $RH_i \geq$ 73



%, but the water needs pores > 9 nm to remain frozen at 259 K because of the melting point depression (see Table 10). Therefore, swelling pores are needed to explain the depositional ice growth observed for oxalic acid. At 268 K, the starting temperature of the expansion run, which led to deliquescence-induced ice growth, the pockets in the oxalic acid particles require diameters of at least 24 nm to maintain the ice. Given the crystallized fraction of 0.13 for deliquescence-induced ice growth from pre-activated particles, such pockets seem to form quite readily when oxalic acid solution droplets effloresce. Ammonium sulfate remained pre-activated when the conditioning temperature was kept below the eutectic temperature but was lost when raising the conditioning temperature above the eutectic temperature, in accordance with pore water being responsible for pre-activation.

### 4.12 AIDA chamber experiments by Wagner et al. (2016)

Wagner et al. (2016) investigated pre-activation by PCF in the AIDA cloud chamber for different INPs. They analyzed the data assuming a two-step pre-activation mechanism involving (i) the capillary condensation of supercooled water below ice saturation, and (ii) the subsequent homogeneous freezing of the capillary-held water without macroscopic ice being produced. Particles with median diameters of about 300 nm were injected into the chamber and cooled overnight to 228 K at 95 % $RH_i$ for pre-activation. Table 11 summarizes the experimental results. Illite NX, diatomaceous earth, two types of zeolites (CBV100 and CBV400), GSG (Graphite Spark Generator) soot, and a natural dust from the Canary Islands (CID) were susceptible to pre-activation, while dust samples from the Sahara (SD2) and Israel (ID), volcanic ash from the Eyjafjallajökull eruption on Iceland in April 2010 (EY01), and water-processed GSG soot did not respond to pre-activation. In all cases, pre-activation was lost when the particles were heated to T > 260 K. For pre-activation by cooling to 228 K at $RH_i$ = 95 % and ice crystal growth up to 259 K at $RH_i$ > 100 %, pore openings with diameters < 6.5 nm are needed to keep pores filled and pore cavities with diameters > 9 nm are needed to preserve ice up to 259 K. These numbers are in agreement with the conclusions by Wagner et al. (2016) that pores with diameters between about 5 and 8 nm contribute to pre-activation under ice-subsaturated conditions.

Wagner et al. (2016) also compared the susceptibility to pre-activation with the intrinsic ability of particles to nucleate an ice cloud, revealing that samples, which responded to pre-activation, produced dense ice clouds during expansion runs performed at temperatures below the homogeneous ice nucleation threshold, when relative humidity exceeded ice saturation. Illite NX exhibited a nucleated fraction of 0.58 at T ≅ 229 K and $RH_i$ < 105 % due to homogeneous freezing of pore water, indicating the presence of pores with diameters between about 3 and 8.5 nm. For the expansion run carried out with pre-activated illite NX, which yielded a pre-activated fraction of 0.06 at ~248 K and $RH_i$ < 105 %, pore diameters > 6 nm are needed to prevent ice from melting in confinement. Therefore, the majority of the pores in illite NX particles are likely between 3 and 6 nm in diameter or even narrower. This is in agreement with results from gas sorption measurements showing that most of the porosity of illites arises from the void spaces between nearly parallel-aligned plates of primary particles resulting in pores of 2 – 5 nm, (Marcolli, 2014; Aylmore, 1974: Aylmore and Quirk, 1967). The relevance of pore width for pre-activation is further supported by the results of the zeolite samples. CBV400 with pores in the range from 4 to 19 nm showed a pre-activated particle fraction





of 0.036 for an expansion run starting at 250 K compared with only < 0.01 for CBV100 with pores in the range from 0.3 nm to 1.2 nm, which are narrower than the critical ice embryo size at this temperature. Therefore, ice cannot form in them.

The pre-activation procedure used by Wagner et al. (2016) is only effective for narrow pores. Pre-activation of wider pores needs higher relative humidity and/or warmer temperatures so that water can condense in them. This explains why samples that

needed $RH_i$ > 110 % to nucleate an ice cloud during expansion runs below the homogeneous ice nucleation temperature were not susceptible to the pre-activation procedure used in this study. If pre-activation had been carried out at warmer temperatures and/or higher humidities, PCF could have occurred also in wider pores. Such a pre-activation procedure might be successful for the volcanic ash sample EY01 which exhibited a nucleated fraction of 0.35 at $T \cong 223$ K and $RH_i$ = 120 %, indicating the presence of pores > 11 nm which are expected to fill and freeze at 120 % $RH_i$.

## 5 Discussion of laboratory studies

### 5.1 Dependence on experimental setup

Two types of setups were used to investigate pre-activation in the laboratory studies summarized in Sect. 4: cloud chambers and cold stages. While the particles are suspended in cloud chambers, they are deposited on substrates in cold stages. In cold stages, gaps and narrow spaces between the substrate and the deposited particles are likely to fill with liquid water below water

saturation due to capillary forces. When the particles are cooled, freezing of this water can lead to pre-activation. Indeed, particles deposited on substrates remained pre-activated after long exposure to dry conditions while in cloud chamber studies, pre-activation was less persistent. Namely, particles on substrates remained pre-activated when exposed to $RH_i \leq 40$ % (Higuchi and Fukuta, 1966; Roberts and Hallett, 1968; Edwards and Evans, 1971; Knopf and Koop, 2006), while in cloud chambers pre-activated particles were observed for less than one minute under such dry conditions (Day, 1958; Mason and Maybank, 1958).

Also, pre-activation proved to be a more common feature for deposited particles than for aerosol particles in cloud chambers. While all substances investigated by Roberts and Hallett (1968) and 12 out of 15 substances investigated by Edward and Evans (1971) showed pre-activation when the particles were deposited on a substrate, only 4 out of 50 substances investigated by Mossop (1956), 2 out of 8 substances investigated by Day (1958) and 10 out of 28 substances investigated by Mason and Maybank (1958) showed pre-activation when the particles were suspended in air. Notably, muscovite showed pre-activation

when deposited (Edwards and Evans, 1971) but not when aerosolized (Mason and Maybank, 1958). Therefore, the results obtained with cold stages are probably biased by pore condensation and freezing of water in voids between the particles and the substrate.



## 5.2 Dependence on particle type

### 5.2.1 Clay minerals

Clay minerals are common components of mineral dusts which are reknowned to nucleate ice heterogeneously (Hoose and Möhler, 2012; Pinti et al., 2012; Murray et al., 2012; Marcolli, 2014). In addition to this intrinsic ice nucleation ability, the studies summarized in Sect. 4 attest clay minerals and clays – soil materials consisting of clay minerals with traces of metal oxides and organic matter – a high susceptibility for pre-activation. In the study by Higuchi and Fukuta (1966), illite, montmorillonite, kaolinite and an unspecified clay showed pre-activation up to 270 – 271 K after having been cooled to 203 K at 40 % $RH_w$. Persistent pre-activation of the clay minerals kaolinite, montmorillonite and Wyoming bentonite – a strongly swelling sodium bentonite – was confirmed by Roberts and Hallett (1968). However, it has to be kept in mind, that Higuchi and Fukuta (1966) and Roberts and Hallett (1968) performed their experiments on cold stages with deposited particles. While the general susceptibility of clay minerals for pre-activation is confirmed by cloud chamber studies, the persistence seems to be less. Mason and Maybank (1958) observed pre-activation for the clay minerals kaolinite, montmorillonite, and sepiolite but only for short exposures of 10 – 30 s to air dried with $CaCl_2$. Wagner et al. (2016) observed pre-activated fractions of 0.006 – 0.06 up to ~256 K for illite when the particles were exposed to $RH_i$ = 83 – 95 % after pre-activation. A strong effect of pre-activation showed montmorillonite and sodium bentonite, which has montomorillonite as a major component. Montmorillonite showed a large increase of the threshold freezing temperature from < 248 K to 263 K (Mason and Maybank, 1958). Sodium bentonite showed a similarly strong increase with a pre-activated fraction of ~0.05 up to 266 K compared with an initial ice-nucleating fraction of 0.001 at 247 K (Mossop, 1956). The ability of montmorillonite, illite, and kaolinite to pre-activate is in accordance with the presence of pores in these particles (Marcolli, 2014; Jeong and Nousiainen, 2014). The large effect of pre-activation of montmorillonite can be explained by the presence of mesopores between primary particles that start to swell well below water saturation (Salles et al., 2009).

On the other hand, halloysite studied by Mason and Maybank (1958) and a clay investigated by Day (1958) did not show pre-activation. Halloysite is a porous clay mineral that can form nanotubes (Churchman et al., 1995; Yuan et al., 2015). It might have failed to show pre-activation if the pores of the investigated halloysite were not the right size for pre-activation. A reason why the clay investigated by Day (1958) was not susceptible to pre-activation might be that the pores were blocked by organic matter, which is a minor additional component of clays, which is not present in the pure clay minerals.

### 5.2.2 Calcium carbonates

$CaCO_3$ is another common component of natural mineral dusts (Usher et al., 2003; Murray et al., 2012), but with a negligible ice nucleation efficiency (Kaufmann et al., 2016; Atkinson et al., 2013). However, it is susceptible to pre-activation. Particles of Iceland spar (a transparent variety of calcite) with ice-nucleating fractions of $10^{-3}$ at 235 K showed pre-activated fractions of 0.01 – 0.05 at T < 269 K and RH below water saturation (Mossop, 1956) after they had been involved in an ice cloud at T ≤ 232 K. However, a precipitated $CaCO_3$ (Analar) exposed to the same procedure did not show pre-activation. Iceland spar that



underwent an ice cloud event at T < 203 K was susceptible to pre-activation in the study by Day (1958) with a pre-activated fraction of 0.01 – 0.05 at 264 K. However, pre-activation vanished within 5 minutes when $RH_i$ was kept between 84 and 98 %. Vaterite, a rare crystal form of $CaCO_3$, started to grow ice at 3 K higher temperature compared with the initial ice nucleation threshold temperature of 266 K when it was pre-activated. In the experiments performed by Roberts and Hallett (1968), calcite

and vaterite both showed a small pre-activated fraction of $10^{-4}$ at 268 K. The porosity of $CaCO_3$ can vary strongly. TEM (Transmission Electron Microscopy) analysis of slices of calcite-rich particles sampled during Asian dust storms revealed the presence of pores of different sizes (10 – 300 nm diameters) and irregular shapes (Jeong and Nousiainen, 2014), while secondary electron images of $CaCO_3$ from the study by Laskin et al. (2006) showed much more compact morphologies.

### 5.2.3 Further mineral dust components

Quartz is a major component of natural mineral dusts (Murray et al., 2012; Kaufmann et al., 2016; Boose et al., 2016). However, quartz particles did not show pre-activation in the studies by Mason and Maybank (1958) and Day (1958). Also α-tridymite and β-tridymite, high-temperature polymorphs of quartz, failed to show pre-activation (Mason and Maybank, 1958). This is in accordance with quartz being a non-porous mineral.

Different feldspars have been investigated by Mason and Maybank (1958). Orthoclase, anorthoclase, and microcline failed to
show pre-activation. Albite showed pre-activation up to 264 K, after having been involved in an ice cloud at T < 255 K. The susceptibility of albite particles to pre-activation was confirmed by the study of Roberts and Hallett (1968), where particles were pre-activated after having been involved in an initial ice cloud event at 250 K. Apparent turbidity of alkali feldspars correlates with the concentrations of micropores, which arise depending on weathering and exhibit sizes from few nanometers to a micrometer, with typical widths of 100 nm. Pristine feldspars, on the other hand, contain only few pores (Walker et al.,
1995). Porosity is also introduced by grinding or when one feldspar is replaced by another one, which is often albite. Albite is typically porous (Hövelmann et al., 2010; Norberg et al., 2011).

Biotite, phlogopite, and muscovite, all members of the mica family, which are sheet silicates with little or no porosity, did not show pre-activation in the cloud chamber study by Mason and Maybank (1958). On the other hand, a mica sample investigated by Higuchi and Fukuta (1966) showed pre-activation. Again, results of this study may be influenced by water present between
the particles and the substrate.

Wagner et al. (2016) investigated three natural dust samples for pre-activation at T = 228 K and $RH_i$ = 95 %. The one from the Canary Island (CID) showed a weak effect of pre-activation. A Saharan dust sample (SD2) and one from Israel (ID) failed to show pre-activation.

### 5.2.4 Volcanic material

Volcanic rock investigated in the cloud chamber by Day (1958) failed to show pre-activation. On the other hand, volcanic ash from Mt. Etna showed pre-activation with a threshold freezing temperature 6 K higher than the initial cloud freezing temperature of 266 K in the cloud chamber experiments by Mason and Maybank (1958). Higuchi and Fukuta (1966) observed high pre-



activated fractions of ~0.7 at 258 K after having cooled a volcanic ash sample below the homogeneous freezing temperature of water. In contrast, Wagner et al. (2016) did not observe enhanced ice formation by a volcanic ash sample from the Eyjafjallajökull eruption, which was pre-activated below ice saturation at T = 228 K. Nevertheless, it might have been susceptible to pre-activation at higher RH, because an expansion run which started at ~227 K yielded an activated fraction of 0.35 at $RH_i$ = 120 %, indicating the presence of larger pores which need higher relative humidity to fill. The porosity of volcanic ash particles is likely to vary depending on the formation conditions, therefore, there should also be variability expected in the susceptibility of particles for pre-activation. Delmelle et al. (2005) investigated the porosity of six different volcanic ash samples. The size distributions they measured peaked at pore diameters of about 5 nm with a long tail to larger diameters.

### 5.2.5 Meteoritic material

Only one of two stony meteorites investigated by Roberts and Hallett (1968) was susceptible to pre-activation with an increase of the threshold freezing temperature by 8 K to 261 K during the second ice cloud event. The stony meteorite investigated by Higuchi and Fukuta (1966) showed a greatly enhanced ice nucleation efficiency at 258 K after pre-activation at temperatures below the homogeneous ice nucleation threshold. Mason and Maybank (1958) investigated three stony meteorites, one showed a threshold freezing temperature of 263 K after pre-activation compared with the initial freezing temperature of 256 K, while the other two failed to show pre-activation. Day (1958) found pre-activation of meteorites when they tested them at 264 K. Similar to the volcanic ash particles, the susceptibility of meteoritic material to pre-activation seems to vary strongly. Meteorites show porosity of varying degree from non-porous to highly porous depending on their composition and their velocity when they entered the atmosphere, which determines the degree of melting they experienced. Porosity of meteorites is often present in the form of spherical vesicles within the meteorites, some of them with openings to the surface (Genge et al. 2008; Taylor et al., 2011; Kohout et al., 2014). Some pores are also of irregular shape. The characterized pores are rather large but smaller pores are also likely to be present. Only the ones with openings to the surface are relevant for pre-activation. Meteorites and therefore also their pores are non-swelling.

### 5.2.6 Substances with a high intrinsic ice nucleation ability

Materials with high intrinsic ice nucleation efficiencies failed to show an additional effect due to pre-activation. No pre-activation was observed for AgI in the studies by Fournier d'Albe (1949), Day (1958), and Roberts and Hallett (1968). Additionally, Edwards and Evans (1971) did not observe pre-activation for $PbI_2$, AgI and CuI, which nucleated ice at 271 K during the first crystallization and also after pre-activation. No effect of pre-activation is expected in the absence of pores or when the investigated substance has a high intrinsic ice nucleation ability that exceeds the pre-activation effect of pores. The closeness of fit between the crystal lattice of an ice-nucleating substrate and ice does not seem to influence the ability of a particle to retain an ice embryo since substances such as AgI and PbI, which have lattice parameters close to those of hexagonal ice did not show any pre-activation (Mossop, 1956).



### 5.3 Dependence on pre-activation conditions

The studies summarized in Sect. 4 show that the main requirement for pre-activation is exposure to subzero temperature. It is not necessary that relative humidity reaches ice saturation (Higuchi and Fukuta, 1966; Wagner et al., 2016) nor is it needed that macroscopic ice forms on the particles. Pre-activation is greatly enhanced when the temperature falls below the homogeneous ice nucleation threshold (Higuchi and Fukuta, 1966) and it is lost for T > 273 K with one exception reported by Edwards et al. (1970). The fraction of particles that remain pre-activated decreases with increasing temperature (Higuchi and Fukuta, 1966; Wagner et al., 2016) and for conditioning at low RH (Day, 1958). All these observations are consistent with pre-activation occurring by a PCF mechanism. The susceptibility to pre-activation correlates well with the porosity of the investigated materials. Nevertheless, this does not preclude that other mechanisms may also be at work. Pre-activation has also been reported for monolayers of long-chain alcohols, which form 2-D crystals on the water surface (Seeley and Seidler, 2001; Zobrist et al., 2007). In this case, a structural rearrangement of the monolayer in response to the ice phase has been proposed as cause for the pre-activation.

Since pre-activation by PCF has bounds to small pore diameters given by the melting point depression in confinement and to large diameters due to evaporation at low RH, swelling pores seem to be best suited for persistent pre-activation. Still, only a small fraction of particles seems to possess pores with the right dimensions to induce ice crystal growth up to high temperatures. The presence of pores generally depends on the inherent porosity of the materials but also on secondary characteristics acquired during particle formation like crystallinity and aggregation. Large particles are more likely aggregates of primary particles with gaps and voids between them suitable for water condensation by capillary forces. Pre-activation at temperatures approaching 273 K is expected for swelling pores, or pockets totally enclosed in particles that dissolve during cloud droplet activation. The modes of freezing for such particles should be condensation and contact freezing, because it occurs only during first contact with liquid water.

### 6 Atmospheric implications

Different scenarios of cloud glaciation in the atmosphere are conceivable for which pre-activation by a PCF mechanism might matter. However, the requirements discussed above have to be fulfilled to preserve ice in pores.

### 6.1 Cloud seeding by pre-activated particles from above

Ice crystals falling from high level ice clouds like cirrus, cumulonimbus anvils, and frontal altostratus clouds are thought to initiate glaciation of lower level clouds or upraising convective clouds by seeding them (Roberts and Hallett, 1968). Ansmann et al. (2009) found that cloud seeding with ice crystals from above (seeder-feeder mechanism) is indeed an important process of ice production in lower layers of multilayer altocumulus systems. Hall and Pruppacher (1976) calculated that ice particles could survive distances of up to 2 km when the relative humidity with respect to ice was below 70 % in a typical mid-latitude atmosphere. If ice is preserved in pores after full sublimation of the macroscopic ice crystals, lower level clouds might be seeded





by pre-activated particles instead of ice crystals (e.g. Knopf and Koop, 2006; Wagner et al., 2014). Relative humidity between ice layers easily falls below 50 % RH$_i$ (Brabec, 2011; Brabec et al., 2012). Because aerosol particles have low terminal velocities, they need to show persistent pre-activation at the humidity conditions of the air masses they pass through. To withstand dry air conditions between cloud layers, ice in swelling pores or ice completely shielded in pockets of effloresced or glassy particles

might be needed (Wagner et al., 2012; 2014).

## 6.2 Wave clouds

Pre-activation might be important for wave clouds which consist of ice crystals and/or water droplets and form directly above or in the lee of a mountain range. Ice in such clouds can form by homogeneous or heterogeneous nucleation (Baker and Lawson, 2006; Field et al., 2001). Heterogeneous ice nucleation in wave clouds in the temperature range from 258 K to 238 K typically

produces ice crystal concentrations from 1 to 10 cm$^{-3}$ (Field et al., 2001). As an air mass passes through multiple waves, it experiences repeated uplift and descent. When the air is humid enough, a cloud is formed during uplift which evaporates again during descent, thus forming a pattern of repeated cloudy and cloud-free regions. Once an ice cloud has formed, a fraction of the involved particles may keep ice in pores and ice crystals may form again on them as soon as ice saturation is exceeded. Orographic lifting and sinking of air masses over mountainous terrain might be well suited for pre-activation when RH falls

only slightly below ice saturation in-between cloudy regions.

## 6.3 Volcanic ash

During volcanic eruptions, ashes are often injected into the high troposphere or even up to the stratosphere where temperatures are low enough for homogeneous ice nucleation of pore water. Particles emitted by volcanoes are in the micrometer size range and settle within hours to days. When relative humidity remains high enough while the particles settle, ice in pores may persist

and induce glaciation by contact nucleation when the particles reach lower level clouds or grow an ice cloud by deposition of water vapor while they pass air layers that are supersaturated with respect to ice.

The recent eruption of the Eyjafjallajökull volcano in southern Iceland in spring 2010 is well documented. Lidar measurements performed by EARLINET (European Aerosol Research Lidar Network; Seifert et al., 2011) over Germany detected at 5 – 6 km height fully glaciated clouds when cloud top temperatures were below 258 K, while under non-volcanic aerosol conditions such

high fractions of fully glaciated clouds were only observed for temperatures below 248 K. On the other hand, ash particles collected at a distance of 58 km from the volcano proved to be virtually inactive as INPs in condensation mode at temperatures above 252 K (Steinke et al., 2011). A small fraction showed activity in immersion mode for temperatures up to 263 K, however, too few to impact ice cloud formation (Hoyle et al., 2011). During the Eyjafjallajökull eruption ash was injected over 9 km high into the atmosphere (Schumann et al., 2011). Particles were often aggregates of several minerals (Hoyle et al., 2011; Schumann

et al., 2011). At these high altitudes, temperatures were low enough for homogeneous freezing of condensed pore water. The largest particles were lost soon due to sedimentation but ash particles with sizes in the micrometer range remained in the atmosphere for several days while gradually settling. When relative humidity remained high enough, the ice is expected to be



preserved within the pores and induce ice clouds. Ash from the Eyjafjallajökull eruption (EY01) did not respond to pre-activation at 228 K and $RH_i = 95$ % in the AIDA chamber study by Wagner et al. (2016), but an expansion run performed at T < 230 K nucleated an ice cloud at $RH_i = 120$ % indicating the presence of pores with diameters > 11 nm, that were too wide to fill at ice-subsaturated conditions (see Sect. 4.12). Ice preserved in such pores could be responsible for the fully glaciated clouds observed for cloud top temperatures below 258 K. This would explain the discrepancy between the observation of fully glaciated clouds at temperatures up to 258 K during the Eyjafjallajökull eruption and the low ice nucleation activity at the same temperature in immersion mode of Eyjafjallajökull particles collected from the ground.

## 6.4 Arctic mixed-phase stratocumuli

Arctic mixed-phase stratocumuli tend to be long lived, with liquid tops that continually precipitate ice. Radiative cooling near cloud top generates turbulence that maintains the liquid-top and forms an approximately well-mixed layer that extends as far as 500 m below cloud base and is frequently decoupled from the surface layer, limiting the flux of aerosols from below (Solomon et al., 2015). Since temperatures are too warm for homogeneous ice nucleation, ice must form heterogeneously. INPs can be entrained into the cloud-driven mixed layer through turbulent mixing from above and/or below. However, measured Arctic INP concentrations are generally much lower than those found at lower latitudes (e.g., Bigg, 1996; Fountain and Ohtake, 1985). During aircraft flights in the Arctic Ocean, Rogers et al. (2001) measured zero up to 100 INPs above water saturation at T = 250 and 253 K with their continuous flow diffusion chamber. Recent studies indicate that entrainment alone cannot account for observed ice crystal number concentration (Fridlind et al., 2012), because INPs should be depleted from the well-mixed boundary layer within minutes. Fridlind et al. (2012) needed to multiply the measured above-cloud INP concentrations by a factor of 30 to reproduce observed ice crystal size distributions. Solomon et al. (2015) investigated the microphysics and dynamics of a cloud-driven mixed layer that was decoupled from surface sources of moisture, heat, and INPs. They examined the role of INP recycling in maintaining ice production using large eddy simulations of a springtime decoupled arctic mixed phase stratocumulus cloud and demonstrated that sustained recycling of INPs through a drying sub-cloud layer and additional activation of INP number concentration due to a cooling cloud layer are sufficient to maintain ice production and that these processes regulate liquid production over multiple days. If INPs are indeed recycled, pre-activation might enhance the ice nucleation activity of porous particles.

Electron microscopy of the INPs collected by Rogers et al. (2001) during aircraft measurements revealed that INPs a few tenths micrometer in size had widely varying morphology and contained crustal materials (primarily Si). Prenni et al. (2009) identified INPs consisting of crustal particles, metal oxides/dust, carbonaceous particles and mixed particles in northern Alaska. If the crustal particles detected by Prenni et al. (2009) included clay minerals, these would be suited for pre-activation with ice crystal growth as soon as RH is above ice saturation. If ice is contained in pockets of carbonaceous particles, relative humidity needs to increase above the deliquescence RH or up to water saturation to release the ice and to initiate ice crystal growth (Wagner et al., 2012; 2014; Adler et al., 2013).





## 6.5 Pre-activation of meteoritic material in the stratosphere

The stratospheric aerosol contains considerable contributions from meteoritic material (Murphy et al., 1998). Curtius et al. (2005) found at altitudes around 18 – 20 km total particle concentrations (0.4 – 20 µm diameters) of ~10 cm$^{-3}$ within and ~20 cm$^{-3}$ outside the polar vortex. About 24% of particles outside the vortex had non-volatile cores. This number raised to 67 % within the polar vortex, most likely due to downward transport from the mesosphere inside the polar vortex. Meteoritic material is mostly internally mixed with sulfuric acid (Murphy et al., 2007). Meteoritic particles have been considered as nucleating particles for nitric acid trihydrate (NAT) (Biermann et al., 1996; Voigt et al., 2005) and ice crystals (Engel et al., 2013; 2014), but have not excelled as very efficient INPs in laboratory studies. However, some of them were susceptible to pre-activation (see Sect. 5.2.5). Pores of meteoritic particles may exhibit various forms including ink-bottle shapes with narrow openings and large cavities. Moreover, when pores are partly filled with sulfuric acid and nitric acid, RH of pore filling and freezing temperature will both be depressed.

Polar stratospheric clouds (PSCs) with varying contributions of STS (supercooled ternary solutions consisting of nitric acid, sulfuric acid and water) particles, NAT and ice form in Arctic and Antarctic winters at typical altitudes of 15 – 25 km. Average temperatures in the polar stratosphere rise to ~225 K in summer and fall below 190 K in winter (Pitts et al., 2011). Assuming a typical value of stratospheric water vapor mixing ratio of 5 ppm, this corresponds to RH$_i$ = 0.2 – 1 % at 225 K and RH$_i$ = 40 – 150 % at 190 K for altitudes of 15 – 25 km. During a polar winter, temperatures typically remain below 200 K, therefore, ice can persist in pores during a whole winter season but does probably not withstand the severe drying in summer. Water is expected to freeze in pores with diameters down to 2.5 nm at temperatures below 200 K. Water fills such narrow pores at 50 % RH$_i$ in the temperature range 190 – 200 K immediately followed by freezing. If cylindrical pores with such narrow diameters were present, pore water would freeze at ice-subsaturated conditions and initiate ice crystal growth as soon as ice saturation is exceeded with no need for pre-activation. However, pore cavities of meteoritic material seem to be rather large although some of them may have narrow openings (see Sect. 5.2.5) and therefore need higher relative humidity to fill. At these low temperatures, even cavities of 6 nm diameters require RH$_i$ > 100 % to fill. If pores are ink-bottle shaped, an energy barrier has to be overcome to empty the pores leading to a hysteresis so that they may remain filled with ice down to low humidity and thus be susceptible to pre-activation.

The Arctic winter 2009/2010 was the focus of the RECONCILE field campaign investigating PSC formation (von Hobe et al., 2013) using balloons and aircraft measurements. Analysis of these measurements was complemented by space-borne lidar measurements from CALIPSO (Pitts et al., 2011). This Arctic winter can be divided into four periods (Pitts et al., 2011). The early season (15 – 30 December 2009) was characterized by low number density liquid/NAT mixtures and no ice clouds followed by a second phase (31 December 2009 – 14 January 2010) with frequent mountain wave ice clouds that nucleated widespread NAT particles (Hoyle et al., 2013). The third phase (15 – 21 January) was characterized by synoptic-scale temperatures below the frost point, which led to an outbreak of widespread ice clouds. The fourth phase (22 – 28 January) marked the end of the PSC season and was characterized by a major stratospheric warming and dominated by PSCs consisting





of STS. For homogeneous ice nucleation on STS particles, temperatures about 3 K below the frost point are needed (e.g. Engel et al., 2013). Such low average temperatures were, however, hardly reached. Detailed microphysical modeling along air parcel trajectories showed that formation of synoptic-scale regions of ice PSCs observed during mid-January 2010 cannot be explained merely by homogeneous ice nucleation but requires heterogeneous nucleation of ice on INPs (Engel et al., 2013). The required

ice nucleation efficiency needed to be comparable to that of e.g. mineral dust particles observed in the troposphere (Engel et al., 2013). However, meteoritic particles did not prove to be efficient INPs in laboratory experiments (e.g. Mason and Maybank, 1958). An explanation for the observed ice clouds could be ice crystal growth on pre-activated meteoritic particles. Pre-activation could have occurred during the first phase of PSC formation with NAT particles. While the temperature at this time of season was too high for ice to form at the surface of the particles, it might have crystallized within the pores of the meteoritic

particles. Alternatively, ice that has formed in the second phase of the PSC season might have survived within pores and initiated ice crystal growth at temperatures just below the frost point.

## 7 Summary and conclusions

The phenomenon of pre-activation was first described by Fournier d'Albe (1949). Since then a number of studies appeared, which are analyzed in this review under the presumption that pre-activation occurred by pore condensation and freezing. Pre-

activation by a PCF mechanism is limited to high temperature by melting of ice in narrow pores and to low relative humidity by sublimation from wide pores, imposing severe restrictions on the pore width range that is susceptible to pre-activation.
The laboratory studies can be divided into cloud chamber and cold stage experiments. Cold stage experiments are performed with deposited particles such that water can gather in voids between the substrate and the particles and cause pre-activation. Indeed, pre-activation persisting at low RH reported for deposited particles in cold stages was not confirmed by cloud chamber

studies with airborne particles. In cloud chamber studies, pre-activated particle fractions are typically $< 0.05$. Low pre-activated fractions are in accordance with a PCF mechanism, relying on the scarce occurrence of pores of the right size and shape. The presence of pores depends on the inherent porosity of the materials but also on secondary characteristics acquired during formation like crystallinity and aggregation. Large particles are more likely aggregates of primary particles with voids between them. The strongest pre-activation effect is expected for swelling pores, or pockets totally enclosed in particles that dissolve

during cloud droplet activation. Such particles should be able to nucleate ice in condensation or contact mode as soon as they come in contact with liquid water. Pores partially filled with condensable material may also show pre-activation. In this case, complete filling occurs at lower RH than for empty pores and the freezing shifts to lower temperatures.
The laboratory studies confirm that particles susceptible to pre-activation are porous. Pre-activation was observed for clay minerals like illite, kaolinite and montmorillonite with inherent porosity. The largest effect was observed for the swelling clay

mineral montmorillonite. Materials that may acquire porosity depending on the formation conditions are $CaCO_3$, meteoritic material and volcanic ash, which showed pre-activation for some samples or in some studies but not in other ones. Some materials like quartz and AgI always failed to show pre-activation. More thorough analysis of the conditions required to preserve





pre-activation is required to confirm and narrow down the theorized dependence on pore size and shape. Moreover, atmospheric ice residuals obtained by drying should be tested for their ice nucleation ability and compared with ice residuals obtained by heating.

Atmospheric relevance of pre-activation by a PCF mechanism may not be generally given but depend on the atmospheric scenario. Lower-level cloud seeding by pre-activated particles released from high-level clouds critically depends on the ability of pores to retain ice at the relative humidity of the air masses they pass through. To judge the potential impact of cloud seeding by pre-activated particles, the relative humidity between cloud layers needs to be assessed. Ice pockets enclosed in deliquescing or dissolving particles or ice in swelling pores have the potential to exhibit persistent pre-activation. However, their atmospheric relevance might be limited by their abundance. Porous particles that are recycled in wave clouds are likely to show pre-activation with ice crystal growth as soon as ice supersaturation is exceeded, once they have undergone an initial ice nucleation event. To confirm this conjecture, wave clouds need to be followed over several freezing-sublimation cycles to investigate whether freezing occurs at higher RH in the first cycle compared with the following ones. Pre-activated volcanic ash particles and meteoritic material are also likely to influence ice cloud formation. The susceptibility of volcanic ash to pre-activation could explain the observation of fully glaciated clouds over Germany at higher temperatures after the eruption of the Eyjafjallajökull than observed over a long time period in the absence of volcanic ash. Moreover, pores in meteoritic particles could be the basis for their ice-nucleating ability. Dedicated modelling studies based on more accurate information on the pre-activation behaviour of the involved particles are needed to further explore these hypotheses.

### Acknowledgments

The author would like to thank Fabian Mahrt and Robert O. David for carefully reading and correcting the manuscript.

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





**Table 1.** Expansion chamber experiments by Fournier d'Albe (1949). $T_{cry}$ – first crystallization temperature; $RH_{cry}$ – first crystallization RH; $F_{cry}$ – crystallized fraction of cloud droplets at $S_w$ or activated fraction of particles below $S_w$; $T_{cond}$ – conditioning temperature; $RH_{cond}$ – conditioning RH; $T_{recry}$ – Recrystallization temperature; $RH_{recry}$ – recrystallization RH with respect to ice; $F_{recry}$ – recrystallized fraction of cloud droplets at $S_w$ or activated fraction of particles below $S_w$; $D_{pore}$ – pore diameter needed to prevent ice from melting; $S_w$ – water saturation; $S_i$ – ice saturation

| INP | $T_{cry}$ | $RH_{cry}$ | $F_{cry}$ | $T_{cond}$ | $RH_{cond}$ | $T_{recry}$ | $RH_{recry}$ | $F_{recry}$ | Pre-act. | $D_{pore}$ |
|---|---|---|---|---|---|---|---|---|---|---|
| $CdI_2$ | ≤ 232 K | $S_w$ | 1 | < 264 K | < $S_i$ | ≤ 263 K | ~111 % | > 0 | yes | > 14 nm |
| NaCl | ≤ 232 K | $S_w$ | 1 | – | < $S_i$ | ≤ 232 K | $S_w$ | 1 | no | – |
| $NaNO_3$ | ≤ 232 K | $S_w$ | 1 | – | < $S_i$ | ≤ 232 K | $S_w$ | 1 | no | – |
| CsI | ≤ 232 K | $S_w$ | 1 | – | < $S_i$ | ≤ 232 K | $S_w$ | 1 | no | – |
| AgI | ≤ 262 K | $S_w$ | > 0 | – | < $S_i$ | ≤ 262 K | $S_w$ | > 0 | no | – |
| AgI | ≤ 256 K | $S_w$ | 1 | – | < $S_i$ | ≤ 256 K | $S_w$ | 1 | no | – |



**Table 2.** Expansion chamber experiments by Mossop (1956). $T_{cry}$ – first crystallization temperature; $RH_{cry}$ – first crystallization RH; $F_{cry}$ – crystallized fraction of cloud droplets at $S_w$ or activated fraction of particles below $S_w$; $T_{cond}$ – conditioning temperature; $RH_{cond}$ – conditioning RH; $T_{recry}$ – recrystallization temperature; $RH_{recry}$ – recrystallization RH with respect to ice; $F_{recry}$ – recrystallized fraction of cloud droplets at $S_w$ or activated fraction of particles below $S_w$; $D_{pore}$ – pore diameter needed to prevent ice from melting. $S_w$ – water saturation. $S_i$ – ice saturation.

| INP | $T_{cry}$ | $RH_{cry}$ | $F_{cry}$ | $T_{cond}$ | $RH_{cond}$ | $T_{recry}$ | $RH_{recry}$ | $F_{recry}$ | Pre-act. | $D_{pore}$ |
|---|---|---|---|---|---|---|---|---|---|---|
| $CdI_2$ (from solution) | 249 K | $S_w$ | ~0.001 | ≤ 264 K | < $S_i$ | < 263 K | < $S_w$ | < 0.001 | yes | > 14 nm |
| $CdI_2$ (from solution) | ≤ 232 K | $S_w$ | 1 | ≤ 264 K | < $S_i$ | 264 K | < $S_w$ | < 0.001 | yes | > 14 nm |
| $CdI_2$ (from solution) | ≤ 232 K | $S_w$ | 1 | ~246 K | < $S_i$ | 245 K | 110 % | 0.05 | yes | > 5 nm |
| $CdI_2$ (from solution) | ≤ 232 K | $S_w$ | 1 | ~259 K | < $S_i$ | 258 K | 107% | 0.05 | yes | > 9 nm |
| $CdI_2$ (smoke) | ≤ 232 K | $S_w$ | 1 | ≤ 261 K | < $S_i$ | 261 K | < $S_w$ | < 0.001 | yes | > 11 nm |
| $CdI_2$ (smoke) | ≤ 232 K | $S_w$ | 1 | ~246 K | < $S_i$ | 245 K | 110 % | 0.05 | yes | > 5 nm |
| $CdI_2$ (smoke) | ≤ 232 K | $S_w$ | 1 | ~259 K | < $S_i$ | 258 K | 107 % | 0.05 | yes | > 9 nm |
| $CaCO_3$ (Iceland spar) | 241 K | $S_w$ | $10^{-4}$ | – | – | – | – | – | – | – |
| $CaCO_3$ (Iceland spar) | 235 K | $S_w$ | $10^{-3}$ | – | – | – | – | – | – | – |
| $CaCO_3$ (Iceland spar) | ≤ 232 K | $S_w$ | 1 | ≤ 269.5 K | < $S_i$ | < 269 K | < $S_w$ | 0.01–0.05 | yes | > 33 nm |
| $CaCO_3$ (Analar) | ≤ 232 K | $S_w$ | 1 | – | < $S_i$ | < 232 K | $S_w$ | 1 | no | – |
| Gypsum | 247 K | $S_w$ | $10^{-4}$ | – | – | – | – | – | – | – |
| Gypsum | 238 K | $S_w$ | $10^{-3}$ | – | – | – | – | – | – | – |
| Gypsum | ≤ 232 K | $S_w$ | 1 | ≤ 267.8 K | < $S_i$ | < 267 K | < $S_w$ | < 0.001 | yes | > 23 nm |
| Na bentonite clay | 247 K | $S_w$ | $10^{-3}$ | – | – | – | – | – | – | – |
| Na bentonite clay | 241 K | $S_w$ | $10^{-2}$ | – | – | – | – | – | – | – |
| Na bentonite clay | ≤ 232 K | $S_w$ | 1 | ≤ 266.6 K | < $S_i$ | < 266 K | < $S_w$ | ~0.05 | yes | > 20 nm |





**Table 3.** Freezing chamber experiments by Day (1958). $T_{cry}$ – first crystallization temperature; $RH_{cry}$ – first crystallization RH; $F_{cry}$ – crystallized fraction of cloud droplets; $T_{cond}$ – conditioning temperature; $RH_{cond}$ – conditioning RH with respect to ice; $t_{cond}$ – conditioning time; $T_{recry}$ – Recrystallization temperature; $F_{recry}$ – recrystallized fraction of cloud droplets; $D_{pore}$ – pore diameter needed to prevent ice from melting; $S_w$ – water saturation.

| INP | $T_{cry}$ | $RH_{cry}$ | $F_{cry}$ | $T_{cond}$ | $RH_{cond}$ | $t_{cond}$ | $T_{recry}$ | $F_{recry}$ | Pre-act. | $D_{pore}$ |
|---|---|---|---|---|---|---|---|---|---|---|
| $CaCO_3$ (Iceland spar) | < 203 K | $S_w$ | 1 | 264 K | 84 – 98 % | 1 min | 264 K | 0.01 – 0.05 | yes | > 14 nm |
| $CaCO_3$ (Iceland spar) | < 203 K | $S_w$ | 1 | 264 K | 84 – 98 % | 3 min | 264 K | $10^{-4}$ – $10^{-3}$ | yes | > 14 nm |
| $CaCO_3$ (Iceland spar) | < 203 K | $S_w$ | 1 | 264 K | 84 – 98 % | 5 min | 264 K | ~$10^{-5}$ | yes | > 14 nm |
| $CaCO_3$ (Iceland spar) | < 203 K | $S_w$ | 1 | 264 K | $CaCl_2$ sat | 1 min | 264 K | 0 | no | – |
| Meteorites | < 203 K | $S_w$ | 1 | 264 K | 84 – 98 % | 1 min | 264 K | ~$10^{-5}$ | yes | > 14 nm |
| Volcanic rock | < 203 K | $S_w$ | 1 | 264 K | 84 – 98 % | 1 min | 264 K | 0 | no | – |
| Quartz | < 203 K | $S_w$ | 1 | 264 K | 84 – 98 % | 1 min | 264 K | 0 | no | – |
| Mica | < 203 K | $S_w$ | 1 | 264 K | 84 – 98 % | 1 min | 264 K | 0 | no | – |
| Clay | < 203 K | $S_w$ | 1 | 264 K | 84 – 98 % | 1 min | 264 K | 0 | no | – |
| Gypsum | < 203 K | $S_w$ | 1 | 264 K | 84 – 98 % | 1 min | 264 K | 0 | no | – |
| AgI | 267 K | $S_w$ | > 0 | – | – | – | – | – | no | – |
| AgI | < 203 K | $S_w$ | 1 | 264 K | 84 – 98 % | 1 min | > 267 K | 0 | no | – |



**Table 4.** Cloud chamber experiments by Mason and Maybank (1958). $T_{cry}$ – threshold temperature of first crystallization with crystallized fraction $F_{cry} \cong 10^{-5}$ at $S_w$; $T_{cond}$ – conditioning temperature.; $RH_{cond}$ – conditioning RH controlled by a saturated CaCl₂ solution, ca 32 % $RH_w$; $t_{cond}$ – conditioning time; $T_{recry}$ – threshold temperature of recrystallization with recrystallized fraction $F_{recry} \cong 10^{-5}$ at $S_w$.

| INP | $T_{cry}$ | $T_{cond}$ | $RH_{cond}$ | $t_{cond}$ | $T_{recry}$ | Pre-act. |
|---|---|---|---|---|---|---|
| Covellite (CuS) | 268 K | 268 K | (CaCl₂)ₛₐₜ | 10 – 30 s | 268 K | no |
| β-Tridymite | 266 K | 268 K | (CaCl₂)ₛₐₜ | 10 – 30 s | 268 K | no |
| Vaterite (CaCO₃) | 266 K | 269 K | (CaCl₂)ₛₐₜ | 10 – 30 s | 269 K | yes |
| Kaolinite | 264 K | 272.6 K | (CaCl₂)ₛₐₜ | 10 – 30 s | 269 K | yes |
| Glacial debris | 263.5 K | 272.1 K | (CaCl₂)ₛₐₜ | 10 – 30 s | 269 K | yes |
| Microcline | 263.5 K | 268 K | (CaCl₂)ₛₐₜ | 10 – 30 s | 263.5 K | no |
| Hematite (Specularite) | 263 K | 268 K | (CaCl₂)ₛₐₜ | 10 – 30 s | 263 K | no |
| Aquadag (colloidal graphite) | 261 K | 268 K | (CaCl₂)ₛₐₜ | 10 – 30 s | 261 K | no |
| Volcanic ash (Mt. Etna) | 260 K | 271.6 K | (CaCl₂)ₛₐₜ | 10 – 30 s | 266 K | yes |
| Halloysite | 260 K | 268 K | (CaCl₂)ₛₐₜ | 10 – 30 s | 260 K | no |
| Dolomite | 259 K | 268 K | (CaCl₂)ₛₐₜ | 10 – 30 s | 259 K. | no |
| Biotite | 259 K | 268 K | (CaCl₂)ₛₐₜ | 10 – 30 s | 259 K | no |
| Vermiculite | 258 K | 268 K | (CaCl₂)ₛₐₜ | 10 – 30 s | 258 K | no |
| Phlogopite | 258 K | 268 K | (CaCl₂)ₛₐₜ | 10 – 30 s | 258 K | no |
| Cinnabar | 257 K | 271 K | (CaCl₂)ₛₐₜ | 10 – 30 s | 266 K | yes |
| Graphite (pencil lead) | 257 K | 272.6 K | (CaCl₂)ₛₐₜ | 10 – 30 s | 264 K | yes |
| Gypsum | 257 K | 268 K | (CaCl₂)ₛₐₜ | 10 – 30 s | 257 K | no |
| One stony meteorite (aerolite) | 256 K | 273.1 K | (CaCl₂)ₛₐₜ | 10 – 30 s | 263 K | yes |
| Anorthoclase | 256 K | 268 K | (CaCl₂)ₛₐₜ | 10 – 30 s | 256 K | no |
| Albite | < 255 K | 270 K | (CaCl₂)ₛₐₜ | 10 – 30 s | 264 K | yes |
| Sepiolite | < 254 K | 270 K | (CaCl₂)ₛₐₜ | 10 – 30 s | 259 K | yes |
| Montmorillonite | < 248 K | 271 K | (CaCl₂)ₛₐₜ | 10 – 30 s | 263 K | yes |
| Muscovite | < 255 K | 268 K | (CaCl₂)ₛₐₜ | 10 – 30 s | < 255 K. | no |
| Orthoclase | < 255 K | 268 K | (CaCl₂)ₛₐₜ | 10 – 30 s | < 255 K | no |
| Talc | < 255 K | 268 K | (CaCl₂)ₛₐₜ | 10 – 30 s | < 255 K | no |
| Sand | < 255 K | 268 K | (CaCl₂)ₛₐₜ | 10 – 30 s | < 255 K | no |
| Quartz | < 255 K | 268 K | (CaCl₂)ₛₐₜ | 10 – 30 s | < 255 K | no |
| Two stony meteorites | < 255 K | 268 K | (CaCl₂)ₛₐₜ | 10 – 30 s | < 255 K | no |
| α-tridymite | < 255 K | 268 K | (CaCl₂)ₛₐₜ | 10 – 30 s | < 255 K | no |



**Table 5**. Particles deposited on a metal foil investigated by Higuchi and Fukuta (1966); $T_{cond}$ – conditioning temperature; $RH_{cond}$ – conditioning RH; $T_{cry}$ – crystallization temperature; $RH_{cry}$ – crystallization RH; $F_{cry}$ – crystallized fraction; $D_{pore}$ – pore diameter needed to prevent ice from melting; $S_w$ – water saturation.

| INP | $T_{cond}$ | $RH_{cond}$ | $T_{cry}$ | $RH_{cry}$ | $F_{cry}$ | Pre-act. | $D_{pore}$ |
|---|---|---|---|---|---|---|---|
| Montmorillonite | 203 K | 40 % | 270 – 271 K | $S_w$ | ~0.001 | yes | > 39 – 56 nm |
| Montmorillonite | 203 K | 40 % | 263 K | $S_w$ | 0.07 – 0.35 | yes | > 13 nm |
| Montmorillonite | 203 K | 40 % | 258 K | $S_w$ | 0.7 – 0.75 | yes | > 9 nm |
| Montmorillonite | 253 K | 40 % | 258 K | $S_w$ | ~0.01 | yes | > 9 nm |
| Montmorillonite | 243 K | 40 % | 258 K | $S_w$ | 0.03 – 0.3 | yes | > 9 nm |
| Montmorillonite | 233 K | 40 % | 258 K | $S_w$ | 0.15 – 0.75 | yes | > 9 nm |
| Montmorillonite | 223 K | 40 % | 258 K | $S_w$ | 0.15 – 0.75 | yes | > 9 nm |
| Montmorillonite | 213 K | 40 % | 258 K | $S_w$ | 0.35 – 0.75 | yes | > 9 nm |
| Volcanic ash | 203 K | 40 % | 270 – 271 K | $S_w$ | ~0.001 | yes | > 39 – 56 nm |
| Volcanic ash | 203 K | 40 % | 268 K | $S_w$ | 0.005 – 0.025 | yes | > 24 nm |
| Volcanic ash | 203 K | 40 % | 263 K | $S_w$ | 0.05 – 0.1 | yes | > 13 nm |
| Volcanic ash | 203 K | 40 % | 258 K | $S_w$ | ~0.7 | yes | > 9 nm |
| Volcanic ash | 253 K | 40 % | 258 K | $S_w$ | ~0.007 | yes | > 9 nm |
| Volcanic ash | 243 K | 40 % | 258 K | $S_w$ | 0.03 – 0.07 | yes | > 9 nm |
| Volcanic ash | 233 K | 40 % | 258 K | $S_w$ | ~0.4 | yes | > 9 nm |
| Volcanic ash | 223 K | 40 % | 258 K | $S_w$ | ~0.7 | yes | > 9 nm |
| Volcanic ash | 213 K | 40 % | 258 K | $S_w$ | ~0.7 | yes | > 9 nm |
| Kaolinite | 203 – 238 K | 40 % | 270 – 271 K | $S_w$ | ~0.001 | yes | > 39 – 56 nm |
| Kaolinite | 203 K | 40 % | 258 K | $S_w$ | ~0.1 | yes | > 9 nm |
| Kaolinite | 203 K | 40 % | 253 K | $S_w$ | ~0.5 | yes | > 7 nm |
| Illite | 203 – 238 K | 40 % | 270 – 271 K | $S_w$ | – | yes | > 39 – 56 nm |
| Mica | 203 – 238 K | 40 % | 270 – 271 K | $S_w$ | – | yes | > 39 – 56 nm |
| Silica gel | 203 – 238 K | 40 % | 270 – 271 K | $S_w$ | – | yes | > 39 – 56 nm |
| Stony meteorite | 203 – 238 K | 40 % | 270 – 271 K | $S_w$ | – | yes | > 39 – 56 nm |
| Clay | 203 – 238 K | 40 % | 270 – 271 K | $S_w$ | – | yes | > 39 – 56 nm |
| Soil | 203 – 238 K | 40 % | 270 – 271 K | $S_w$ | – | yes | > 39 – 56 nm |



**Table 6.** Particles deposited on glass cover slips investigated by Roberts and Hallett (1968). $T_{cry}$ –crystallization temperature at $S_w$ for nucleated fractions of $F = 10^{-4}$ and $F = 10^{-2}$; $T_{cry}$ / $RH_{cry}$ – threshold crystallization temperature and RH with respect to ice for nucleation below $S_w$; $RH_{cond}$ – conditioning RH with respect to ice; $t_{cond}$ – conditioning time; $T_{recry}$ / $RH_{recry}$ – threshold recrystallization temperature and RH with respect to ice for nucleation below $S_w$

| INP | $T_{cry,}$ $F = 10^{-4}$ | $T_{cry,}$ $F = 10^{-2}$ | $T_{cry,}$ / $RH_{cry}$ | $RH_{cond}$ | $t_{cond}$ | $T_{recry,}$ $F = 10^{-4}$ | $T_{recry,}$ $F = 10^{-2}$ | $T_{recry,}$ / $RH_{recry}$ |
|---|---|---|---|---|---|---|---|---|
| Kaolinite | 262.5 K | 258 K | 254 K / 120% | 35 % | days | 269 K | 266 K | 261.5 K/112% |
| Montmorillonite | 248 K | 246 K | < 246 K / – | 20 % | days | 269 K | 268 K | 259.5 K/114 % |
| Wyoming bentonite | 247 K | 245 K | < 245 K / – | 20 % | days | – | 268 K | 259.5 K/114 % |
| Surface glacier debris: Blue Glacier Washington | 256.5 K | – | 254 K/120 % | 35 % | days | 266 K | – | 258.5 K/115 % |
| Surface glacier debris: Alfotbreen | 258.5 K | – | 256 K /118 % | 35 % | days | 267 K | – | 261.5 K/112 % |
| Surface glacier debris: Gorner Glacier | 266 K | 261 K | 252.5 K /122 % | 40 % | days | 268.5 K | – | 258 K/116 % |
| Stony meteorite | 253 K | – | – / – | 40 % | days | 261 K | – | 256 K 118 % |
| Stony meteorite | 249 K | – | – / – | – | | no pre-act. | | – / – |
| Gypsum | 257 K | 252 K | 254 K /120 % | 30 % | days | 268 K | 265 K | 258 K/116 % |
| Calcite (CaCO₃) | 255 K | 253 K | 254 K /120 % | 60 % | <5 min | 268 K | 261 K | 258.5 K/115 % |
| Vaterite (CaCO₃) | 261 K | 258 K | – | – | – | 268 K | – | 258.5 K/115 % |
| Albite | 250 K | – | – | 40 % | days | 262.5 K | – | 258.5 K/115 % |





**Table 7.** Particles deposited on hydrophobic glass cover slips investigated by Edwards and Evans (1971). $T_{cry}$ – crystallization temperature to obtain a nucleated fraction of $F = 10^{-3}$ after 5 min at $RH_w = 120\ \%$. $T_{recry}(w)$– recrystallization temperature to obtain a nucleated fraction of $F = 10^{-3}$ after 5 min at $RH_w = 120\ \%$ after activation at $RH_w = 120\ \%$ and T = 243 K followed by drying at $RH_{cond}$ with respect to ice for a time $t_{cond}$; $T_{recry}(d)$– recrystallization temperature to obtain a nucleated fraction of $F = 10^{-3}$ after 5 min at $RH_w = 120\ \%$ after activation at $RH_w = 80\ \%$ and T = 243 K.

| INP | $T_{cry}$ | $T_{recry}$ (w) | | | | $T_{recry}$ (d) | *Pre-act.* |
| | | $RH_{cond} = 80\ \%$ | | $RH_{cond} = 40\ \%$ | | | |
| | | $t_{cond} = 2\ min$ | $t_{cond} = 45\ min$ | $t_{cond} = 2\ min$ | $t_{cond} = 45\ min$ | | |
| --- | --- | --- | --- | --- | --- | --- | --- |
| HgI$_2$ | 264 K | 271 K | 271 K | – | – | 271 K | yes |
| PbI$_2$ | 271 K | 271 K | – | – | – | – | no |
| Gypsum | 260 K | 272 K | 272 K | 268 K | – | – | yes |
| Muscovite | 255 K | 270 K | 270 K | < 267 K | – | – | yes |
| AgI | 271 K | 271 K | – | – | – | – | no |
| CuI | 271 K | 271 K | – | – | – | – | no |
| Silica gel | 263 K | 268 K | 268 K | 267 K | 266 K | 268 K | yes |
| Calcite (CaCO$_3$) | 263 K | 268 K | 268 K | 265 K | – | – | yes |
| Kaolinite | 260 K | 271 K | 271 K | – | – | – | yes |
| Alumina | 253 K | 269 K | 269 K | – | – | – | yes |
| Phloroglucinol dihydrate | 268 K | 272 K | 272 K | 272 K | 272 K | 272 K | yes |
| α-Phenazine | 269 K | 272 K | – | – | – | – | yes |
| l-asparagine | 270 K | 272 K | – | – | – | – | yes |
| Egg albumin | 260 K | 270 K | – | – | – | – | yes |
| Benzil | 269 K | 271 K | – | – | – | – | yes |



**Table 8.** Particles deposited on hydrophobic substrates by Knopf and Koop (2006). $T_{cry}$ –crystallization temperature; $RH_{cry}$ – crystallization RH with respect to ice; $T_{cond}$ – conditioning temperature; $RH_{cond}$ – conditioning RH with respect to ice; $T_{recry}$ – recrystallization temperature; $\Delta RH_{recry}$ – difference between crystallization and recrystallization RH with respect to ice; $S_w$ – water saturation.

| INP | $T_{cry}$ | $RH_{cry}$ | $T_{cond}$ | $RH_{cond}$ | $T_{recry}$ | $\Delta RH_{recry}$ |
|-----|-----------|------------|------------|-------------|-------------|---------------------|
| ATD | 203 K | 100 – 150 % | 203 K | 5 – 40 % | 203 K | 0 – 32 % |
| ATD | 207 K | 105 – 170 % | 207 K | 5 – 40 % | 207 K | 0 – 36 % |
| ATD | 213 K | 130 – 160 % | 213 K | 5 – 40 % | 213 K | 0 – 27 % |
| ATD | 218 K | 100 – 150 % | 218 K | 5 – 40 % | 218 K | 0 – 35 % |
| ATD | 230 K | 100 – 150 % | 230 K | 5 – 40 % | 230 K | 16 – 45 % |
| ATD | 239 K | 100 % – $S_w$ | 239 K | 5 – 40 % | 239 K | 0 – 15 % |
| ATD | 250 K | 107 % – $S_w$ | 250 K | 5 – 40 % | 250 K | 1 – 18 % |
| ATD | 260 K | 105 % – $S_w$ | 260 K | 5 – 40 % | 260 K | 1 – 15 % |

**Table 9.** AIDA chamber experiments by Wagner et al. (2012). $T_{cry}$ – crystallization temperature; $RH_{cry}$ – crystallization RH with respect to ice; $C_{cry}$ – ice crystal concentration during first expansion run; $T_{cond}$ – conditioning temperature with respect to ice; $RH_{cond}$ – conditioning RH with respect to ice; $t_{cond}$ – conditioning time; $T_{recry}$ – recrystallization temperature; $RH_{recry}$ – recrystallization RH with respect to ice; $C_{recry}$ –ice crystal concentration during second expansion run. *particles passed glass

10 transition from liquid to glassy while freezing homogeneously during first expansion. **freeze concentrated solution is highly viscous after homogeneous nucleation during first expansion. *** Particles remained liquid during homogeneous freezing in the first expansion run.

| INP | $T_{cry}$ | $RH_{cry}$ | $C_{cry}$ | $T_{cond}$ | $RH_{cond}$ | $t_{cond}$ | $T_{recry}$ | $RH_{recry}$ | $C_{recry}$ | Pre-act. |
|-----|-----------|------------|-----------|------------|-------------|------------|-------------|--------------|-------------|----------|
| Raffinose* | ~230 K | ~138 % | ~100 cm$^{-3}$ | ~224 K | 70 – 80 % | ~ 2.5 h | ~221 K | ~112 % | 20 cm$^{-3}$ | yes |
| Raffinose* | ~226 K | ~138 % | ~100 cm$^{-3}$ | ~230 K | 70 – 80 % | ~15 min | ~229 K | ~105 % | 35 cm$^{-3}$ | yes |
| HMMA* | ~228 K | ~138 % | ~170 cm$^{-3}$ | ~232 K | ≥ 72 % | ~15 min | ~231 K | ~105 % | 30 cm$^{-3}$ | yes |
| Raffinose/M5AS** | ~211 K | ~150 % | ~100 cm$^{-3}$ | ~216 K | 70 – 80 % | ~15 min | ~212 K | ~130 % | ~3 cm$^{-3}$ | yes |
| Raffinose/M5AS*** | ~219 K | ~145 % | ~100 cm$^{-3}$ | ~215 K | 70 – 80 % | ~15 min | ~211 K | ~160 % | ~100 cm$^{-3}$ | no |





**Table 10.** AIDA chamber experiments by Wagner et al. (2014). $T_{liquid}$ – temperature of liquid cloud activation; $T_{cond}$ – conditioning RH with respect to ice; $RH_{cond}$ – conditioning RH with respect to ice; $t_{cond}$ – conditioning time; $T_{start}$ – starting temperature of the expansion run; $T_{cry}$ – crystallization temperature of pre-activated particles; $RH_{cry}$ – crystallization RH with respect to ice; $F_{preact}$ – pre-activated fraction of particles; $D_{pore}$ – pore diameter needed to prevent ice from melting; AS – ammonium sulfate; DRH – deliquescence RH; ET – eutectic temperature.

| INP | $T_{liquid}$ | $T_{cond}$ | $RH_{cond}$ | $t_{cond}$ | $T_{start}$ | $T_{cry}$ | $RH_{cry}$ | $F_{preact}$ | Pre-act. | $D_{pore}$ |
|---|---|---|---|---|---|---|---|---|---|---|
| Succinic acid | ~246 K | ≥ 227.3 K | ≥ 72 % | ~17 h | ~248 K | ~246 K | ~112 % | 0.04 | yes | > 5 nm |
| Succinic acid | ~246 K | ≥ 227.3 K | ≥ 72 % | ~17 h | ~265.5 K | ~264 K | $S_w$ | 0.2 | yes | > 16 nm |
| Oxalic acid | ~256 K | 244 K | ≥ 73 % | hours | ~259 K | ~256 K | 103 % | 0.03 | yes | > 9 nm |
| Oxalic acid | ~256 K | 244 K | ≥ 73 % | hours | ~260 K | ~259 K | 102 % | 0.04 | yes | > 10 nm |
| Oxalic acid | ~256 K | 244 K | ≥ 73 % | hours | ~268 K | ~267 K | $S_w$ | 0.13 | yes | > 24 nm |
| AS | ~240 K | ≥ 196 K | ≥ 66 % | hours | ~245 K | ~244 K | 102 % | 0.01 | yes | – |
| AS | ~240 K | ≥ 196 K | ≥ 66 % | hours | ~251 K | ~249 K | > DRH | 0.04 | yes | – |
| AS | ~240 K | 1.≥ 196 K, 2. > ET | ≥ 66 % | hours | ~245 K | – | > DRH | ~ 0 | no | – |

**Table 11.** AIDA chamber experiments by Wagner et al. (2016). $T_{preact}$ – pre-activation temperature; $RH_{preact}$ pre-activation RH with respect to ice; $RH_{start}$ – RH with respect to ice at the start of the expansion run; $T_{start}$ – starting temperature of the expansion run; $T_{cry}$ –crystallization temperature; $RH_{cry}$ – crystallization RH with respect to ice; $F_{cry}$ – crystallized particle fraction; $D_{pore}$ – pore diameter needed to prevent ice from melting; $S_w$ – water saturation.

| INP | $T_{preact}$ | $RH_{preact}$ | $RH_{start}$ | $T_{start}$ | $T_{cry}$ | $RH_{cry}$ | $F_{cry}$ | Pre-act. | $D_{pore}$ |
|---|---|---|---|---|---|---|---|---|---|
| Zeolite CBV400 | – | – | 88 % | ~251 K | ~246 K | $S_w$ | 0.001 | no | – |
| Zeolite CBV400 | – | – | 95 % | ~228 K | ~227 K | 102 % | 0.4 | no | – |
| Zeolite CBV400 | 228 K | 95 % | 92 % | ~250 K | ~249 K | 102 % | 0.036 | yes | > 6 nm |
| Zeolite CBV400 | 228 K | 95 % | – | ~253 K | – | < $S_w$ | 0.017 | yes | > 7 nm |
| Zeolite CBV400 | 228 K | 95 % | – | ~256 K | – | < $S_w$ | 0.002 | yes | > 8 nm |
| Zeolite CBV400 | 228 K | 95 % | 90 % | ~259 K | ~256 K | $S_w$ | 0.001 | yes | > 9 nm |
| Zeolite CBV100 | 228 K | 95 % | – | ~246 K | – | < $S_w$ | 0.01 | Yes | > 5 nm |
| Zeolite CBV100 | 228 K | 95 % | 95 % | ~250 K | ~248 K | < $S_w$ | 0.001 | yes | > 6 nm |
| Zeolite CBV100 | 228 K | 95 % | 92 % | ~250 K | ~246 K | $S_w$ | 0.006 | yes | > 6 nm |
| Zeolite CBV100 | 228 K | 95 % | 90 % | ~253 K | – | $S_w$ | ~0 | no | > 7 nm |
| Illite NX | – | – | 98 % | ~250 K | ~243 K | $S_w$ | < 0.01 | no | – |
| Illite NX | – | – | 98 % | ~250 K | ~242 K | $S_w$ | 0.04 | no | – |
| Illite NX | – | – | 98 % | ~230 K | ~229 K | < 105 % | 0.58 | no | – |





| | | | | | | | | | |
|---|---|---|---|---|---|---|---|---|---|
| Illite NX | 228 K | 95 % | 86 % | ~250 K | ~248 K | < 105 % | 0.06 | yes | > 6 nm |
| Illite NX | 228 K | 95 % | 83 % | ~253 K | ~250 K | < 105 % | 0.03 | yes | > 7 nm |
| Illite NX | 228 K | 95 % | – | ~256 K | – | < $S_w$ | 0.006 | yes | > 8 nm |
| Illite NX | 228 K | 95 % | – | ~259 K | – | $S_w$ | 0.004 | yes | > 9 nm |
| Diatomaceous earth | – | – | 98 % | ~250 K | ~243 K | $S_w$ | 0.001 | no | – |
| Diatomaceous earth | – | – | 98 % | ~250 K | ~241 K | $S_w$ | 0.01 | no | – |
| Diatomaceous earth | – | – | 95 % | ~245 K | ~239 K | $S_w$ | 0.7 | no | – |
| Diatomaceous earth | – | – | 92 % | ~233 K | ~231 K | 104 % | 0.8 | no | – |
| Diatomaceous earth | 228 K | 95 % | 81 % | ~256 K | ~253 K | 105 % | 0.002 | yes | > 8 nm |
| Diatomaceous earth | 228 K | 95 % | 81 % | ~256 K | ~252 K | 110 % | 0.01 | yes | > 8 nm |
| Diatomaceous earth | 228 K | 95 % | – | ~250 K | – | < $S_w$ | 0.038 | yes | > 6 nm |
| Diatomaceous earth | 228 K | 95 % | – | ~253 K | – | < $S_w$ | 0.02 | yes | > 7 nm |
| Diatomaceous earth | 228 K | 95 % | – | ~256 K | – | < $S_w$ | 0.01 | yes | > 8 nm |
| Diatomaceous earth | 228 K | 95 % | – | ~259 K | – | < $S_w$ | 0.002 | yes | > 9 nm |
| Diatomaceous earth | 228 K | 95 % | – | ~259 K | – | $S_w$ | 0.008 | yes | > 9 nm |
| GSG soot | – | – | 98 % | ~246 K | <239 K | $S_w$ | <0.001 | no | – |
| GSG soot | 228 K | 95 % | 98 % | ~246 K | ~244 K | 110 % | 0.001 | yes | > 5 nm |
| GSG soot | 228 K | 95 % | 98 % | ~246 K | ~243 K | 115 % | 0.002 | yes | > 5 nm |
| GSG soot | 228 K | 95 % | – | ~250 K | – | $S_w$ | 0.001 | no | – |
| Canary Island dust, CID | – | – | 95 % | ~228 K | ~225 K | < 110 % | 0.6 | no | – |
| Canary Island dust, CID | 228 K | 95 % | 88 % | ~250 K | ~247.5 K | 105 % | 0.001 | yes | > 6 nm |
| Canary Island dust, CID | 228 K | 95 % | 88 % | ~250 K | ~246 K | 120 % | 0.005 | yes | > 6 nm |
| Canary Island dust, CID | 228 K | 95 % | 88 % | ~250 K | ~244 K | 120 % | 0.03 | yes | > 6 nm |
| Sahara dust, SD2 | – | – | 92 % | ~224 K | ~222 K | 130 % | 0.15 | no | – |
| Sahara dust, SD2 | 228 K | 95 % | n.g. | n.g. | n.g. | n.g. | n.g. | no | – |
| Israeli dust, ID | – | – | 91 % | ~228 K | ~226 K | 120 % | 0.05 | no | – |
| Israeli dust, ID | 228 K | 95 % | n.g. | n.g. | n.g. | n.g. | n.g. | no | – |
| Volcanic ash, EY01 | – | – | 88 % | ~227 K | ~223 K | 120 % | 0.35 | no | – |
| Volcanic ash, EY01 | 228 K | – | n.g. | n.g. | n.g. | n.g. | n.g. | no | – |
| Water processed GSG soot | 228 K | – | n.g. | n.g. | n.g. | n.g. | n.g. | no | – |



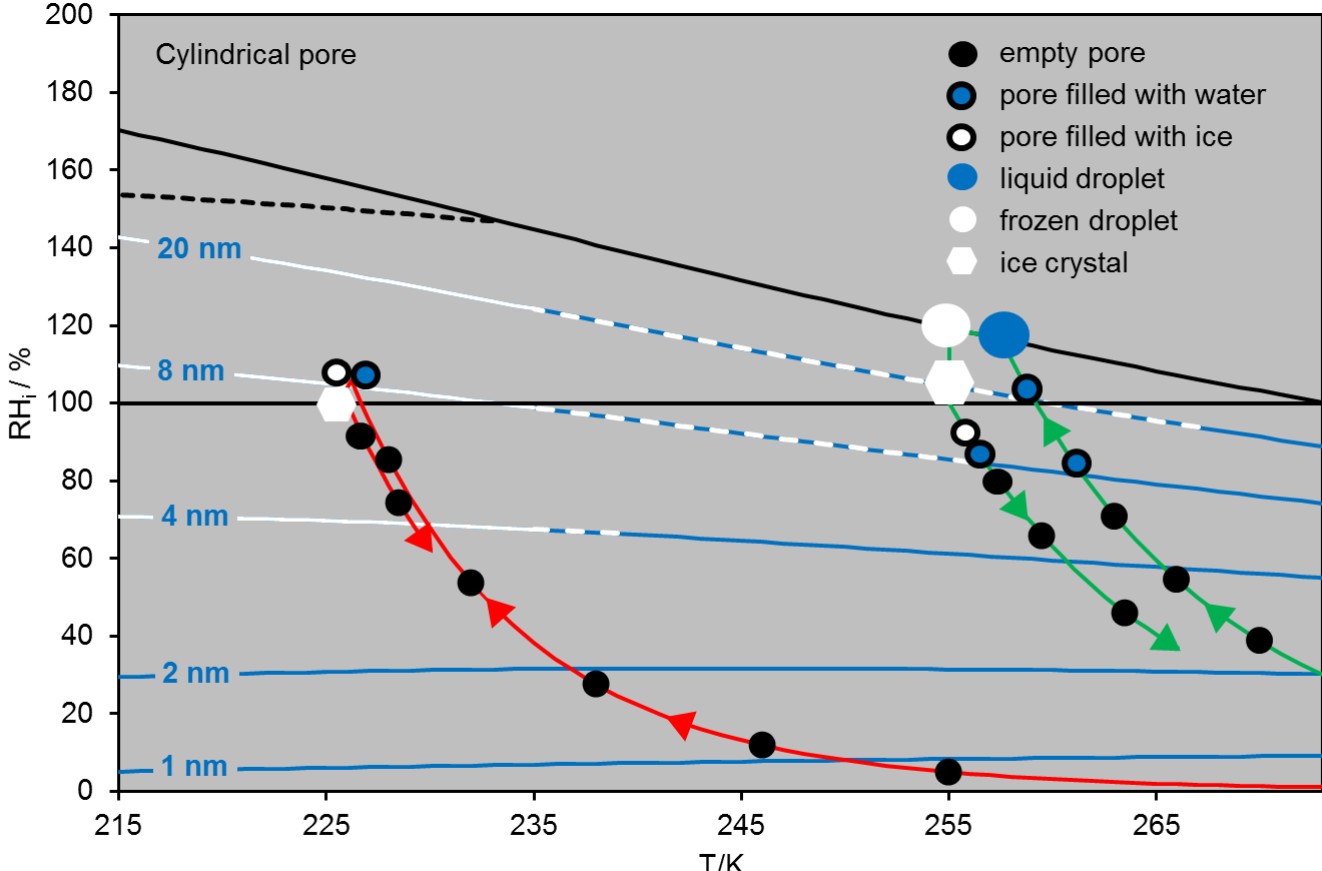

**Figure 1.** Wet trajectory (green line) and dry trajectory (red line) of a particle with a cylindrical pore of 8 nm diameter. Adiabatic cooling, followed by adiabatic heating. The black horizontal line denotes ice saturation; the black sloped line indicates $RH_i/T$ conditions for water saturation (parameterization of Murphy and Koop, 2005). The black dashed line gives homogeneous ice nucleation according to Koop and Zobrist (2009). The white/blue lines delimit the onset of pore filling, which were calculated using the inverse Kelvin equation (Eq. 1). Pores with diameters given on the lines are filled at $RH_i$ values above the line and empty below the line. The white portion denotes pore ice, the blue one pore water, and the dashed portion pore water or pore ice depending on the particle history.





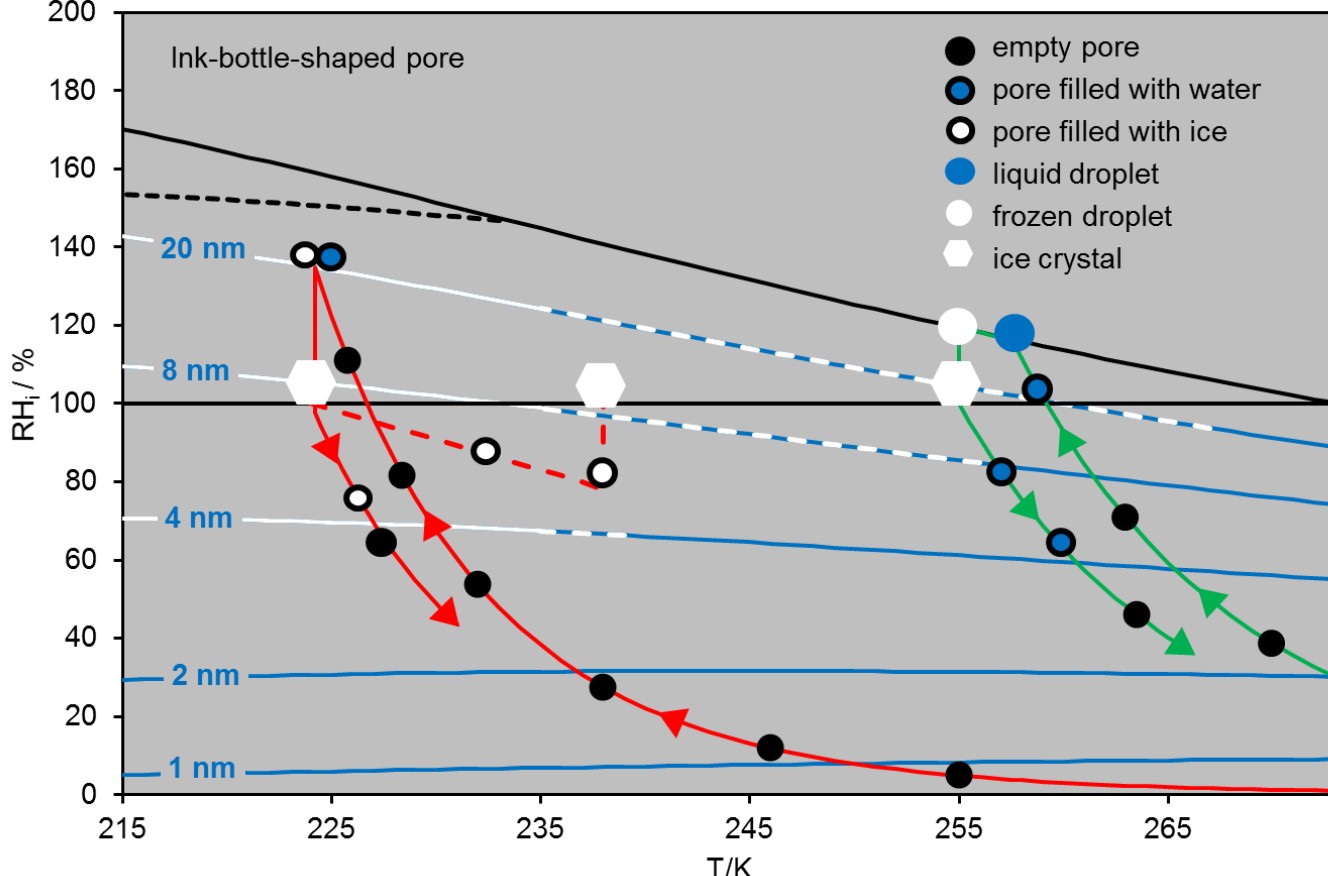

**Figure 2.** Wet trajectory (green line) and dry trajectory (red line) of a particle with an ink-bottle-shaped pore with a cavity of 20 nm and a pore opening of 4 nm diameter. Solid lines: Adiabatic cooling, followed by adiabatic heating; dashed line: heating to 241 K while keeping $RH_i > 70$ %. The black horizontal line denotes ice saturation; the black sloped line indicates $RH_i/T$ conditions for water saturation (parameterization of Murphy and Koop, 2005). The black dashed line gives homogeneous ice nucleation according to Koop and Zobrist (2009). The white/blue lines delimit the onset of pore filling, which were calculated using the inverse Kelvin equation (Eq. 1). Pores with diameters given on the lines are filled at $RH_i$ values above the line and empty below the line. The white portion denotes pore ice, the blue one pore water, and the dashed portion pore water or pore ice depending on the particle history.





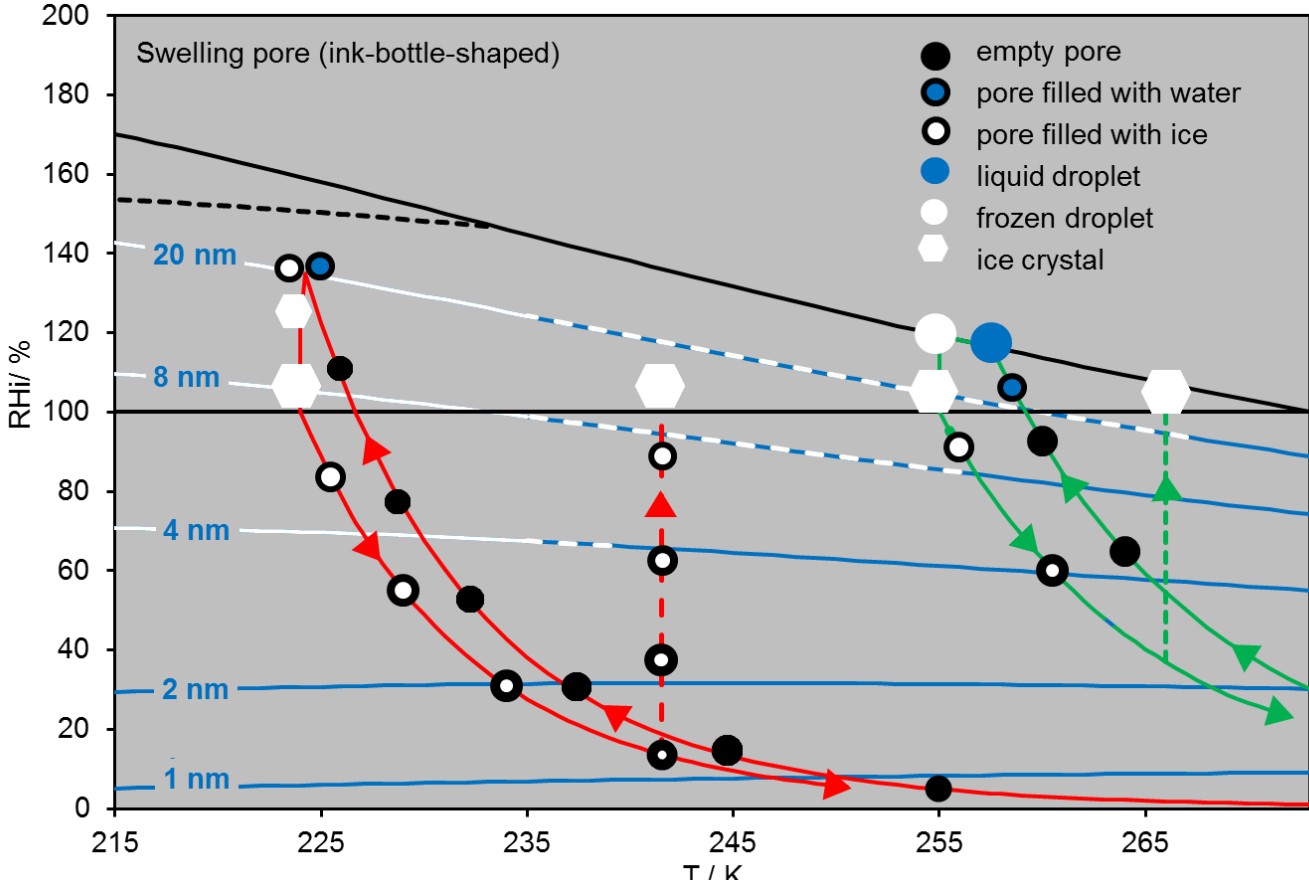

**Figure 3.** Wet trajectory (green line) and dry trajectory (red line) of a particle with a swelling ink-bottle shaped pore with a cavity of 20 nm and a pore opening that reacts to RH changes. Solid lines: Adiabatic cooling, followed by adiabatic heating; dashed lines: increase of RH at constant temperature. The black horizontal line denotes ice saturation; the black sloped line indicates $RH_i/T$ conditions for water saturation (parameterization of Murphy and Koop, 2005). The black dashed line gives homogeneous ice nucleation according to Koop and Zobrist (2009). The white/blue lines delimit the onset of pore filling, which were calculated using the inverse Kelvin equation (Eq. 1). Pores with diameters given on the lines are filled at $RH_i$ values above the line and empty below the line. The white portion denotes pore ice, the blue one pore water, and the dashed portion pore water or pore ice depending on the particle history.



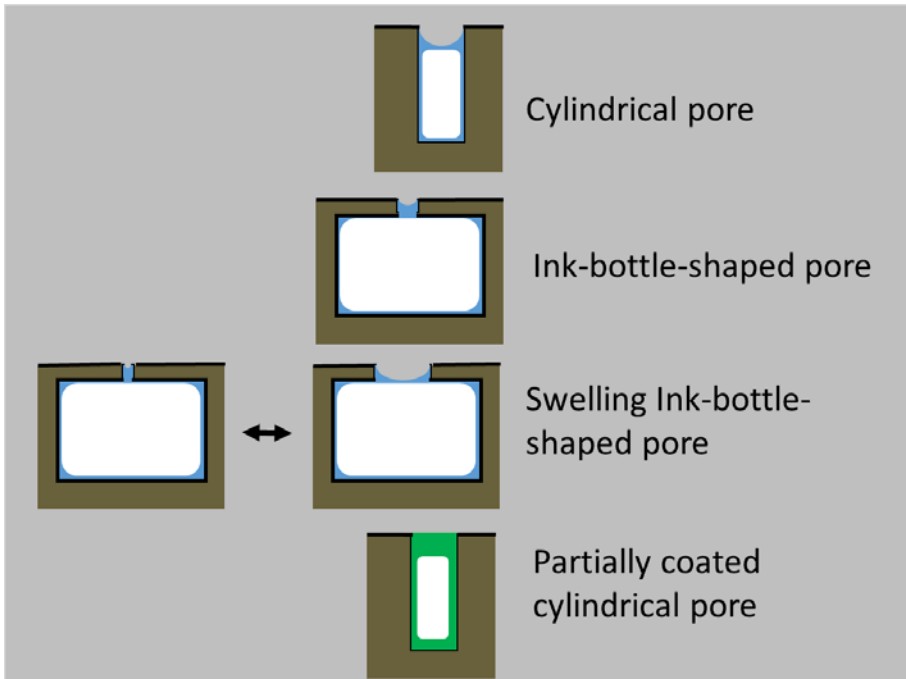

**Figure 4.** Pore types investigated for their capability to pre-activate. The pores are sketched in pre-activated state with ice inside. Colour code: white – ice; blue – water; green – concentrated solution; brown – pore wall. Note that at the wall and in the opening, a quasi-liquid layer of water or in case of the coated pore of concentrated solution is present.

