# Peer review of "Pre-activation of aerosol particles by ice preserved in pores"

_Atmospheric Chemistry and Physics, 2016_

## Referee Comment (RC1) · Anonymous Referee #1 · 18 Oct 2016

This manuscript puts forward a literature review-based concept of pre-activation by aerosol particles via the pore condensation and freezing (PCF) mechanism. Application of this mechanism suggests that pre-activation by PCF is constrained by the melting of ice in narrow pores and the sublimation of ice from wide pores. For these reasons, the author argues pre-activation for cylindrical pores is imposed by restrictions on the temperature and relative humidity range. In addition to reviewing previous experimental data sets with regard of finding indications of this concept, the author also puts forward atmospheric scenarios where pre-activation may play a significant role in atmospheric ice formation.

The topic of this manuscript fits within the scope of ACP. The author carefully reviewed the previous literature dealing with pre-activation phenomena. Although, I like the proposed concept and the effort to use previous data for interpretation, I feel some revisions that deal with the general uncertainty of proposed concept and data, are necessary before this manuscript can be published. The author has my full support of publishing this manuscript, hopefully encouraging further experimental investigation of this effect.

As written, the manuscript often reads as if the novel concept is a "fact". One has to keep in mind that there is no experimental in situ proof of the suggested mechanisms for discussed and investigated particles. Considering this, some statements appear "too factual" and thus should be changed in a way to convey the suggestive nature of this discussion.

For example, the ice formation experiments from the second half of the last century are not well constrained in terms of particle and ice crystal numbers, relative humidity, etc. Often no control or calibration experiments were performed. Considering that even current ice nucleation experiments deviate significantly (see recent data reviews or intercomparison studies), the experimental data can very likely not be used as a definitive support of the proposed concept. This is also indicated by the values in the presented tables which do not include any uncertainties and in many cases the errors, I believe, cannot even be defined or are just very large. Keeping this all in mind, some statements should be more adequately formulated.

For this review, I read Marcolli (2014) that introduces PCF. It is argued that homogeneous freezing occurs in the nanometer-sized pores. From this, as far as I understand, the critical size of the ice embryo fitting inside a pore is derived. However, does homogeneous freezing not also depend on the volume and time? The homogeneous freezing line corresponds to about $J_{hom}=1E10$ cm-3 s-1 (Koop et al., 2000). Pores 4 – 20 nm wide and about 16-20 nm deep have a volume of about 1E-19 cm3, resulting in an ice nucleation rate of about 1E-9 s-1. Obviously, one would need to wait 1E9 s at those fixed conditions to observe 1 ice nucleation event in 1 second. The liquid in 1E18 pores would be needed to observe a freezing event in 1 second. Maybe Jhom in pores is different but then other aspects/assumptions break down. Very recently Koop and

Murray (2016) showed that Jhom is not continuously increasing with decreasing temperature, limiting the rate for nucleation to about 1E12 cm-3 s-1. Maybe I am missing here something? My point is that all reported or applied ice nucleation data sets inherently are based on different particle surface areas and experimental time scales and have different pore numbers (and sizes), all of which are mostly unknown or associated with large uncertainties. Thus, it is very unlikely that any of the stated experiments can be used to make a definitive case for pre-activation by PCF.

The same discussion/exercise can be done assuming immersion freezing in a pore by an active site. Immersion freezing and deposition ice nucleation are known to depend on particle surface area (e.g. Kanji et al., 2008). Looking at the literature (e.g. review article by Murray et al. 2012) it looks like "a lot of surface area" has to be provided to detect ice formation. For example, typical experimental particulate surface areas are larger than 1E14 nm2 to observe ice formation. Many pores are needed that contain an active site to be able to reproduce the data sets.

I am not stating this to cast doubt on the PCF mechanism, which I like and support, but at current stage I recommend to be more careful how to discuss this concept with regard to experimental data. Having said all that, I am not surprised to see some experiments somehow following the presented concept and some not, even if same or similar porous materials were applied. The data sets are just not sufficiently constrained. Statements that a particular approach, such as the cold stage experiment, as discussed in more detail below, is producing potentially erroneous data with respect to pre-activation is, however, unfounded and should be discarded. With present uncertainties and lack of experimental proof, those statements are unjustified. As a matter of fact, these statements detract from the overall nice manuscript.

Page 5-6, section 3: It would be interesting to know how long it takes for ice or water to evaporate from the different pores. This could be done as a function of difference of pore equilibrium RH and ambient RH (and exemplary pore size). This would give an idea if the transient state is important or not. In particular, in an actual cloud with

eddies (up/downdraft), the transient state may be a crucial parameter.

Page 7, line 10: "However, . . .". This sentence seems to be confusing.

Page 8, line 25: "A freeze concentrated. . .". How are the water activity values derived?

Page 9, line 28: I highly doubt that the freezing point in that type of experiment can be measured to this degree in 1949. This may not be even possible today.

Page 10, line 28: "Results of ball milled Iceland spar in the size range from 1 – 15 $\mu$m with large numbers from 1 – 3 $\mu$m were presented in most detail: 1 – 5 % of the particles showed pre-activation when kept for 1 min at 84 – 98 % RHi (see Table 3)." This sounds a bit confusing: Did you mean "Results of ball milled Iceland spar particles, in the size range from 1 – 15 $\mu$m with the largest particle numbers in the size from 1 – 3 $\mu$m, were discussed/investigated in most detailine In this case, 1 – 5 % of the particles showed pre-activation when kept for 1 min at 84 – 98 % RHi (see Table 3)."?

Page 11, line 22-24: Can it be shown quantitatively that equilibrium was not reached? This is related to my comment above regarding sublimating ice.

Page 12, line 33: "However, . . .". Please avoid this statement. There is no evidence for this and just speculation. Though the authors of this study did not use microscopic techniques, as far as I recall this work, this is just not a qualified statement. With better experiments in the future, time will tell. One cannot just say a technique is "wrong" when it does not "obey" a new concept.

Page 13, line 18: "Therefore,. . .". Again this is an unsubstantiated statement considering all uncertainties and should be omitted. In fact, Roberts and Hallet observed the particles and ice crystals with a microscope. Some general remarks for this study and following cold stage experiments below:

If ice forms between a particle and substrate, it will move the particle and the sample image would change. Any microscopist would observe and notice this effect and this would have been long established in the community. This is so significant that it would

have not been missed. In particular, when looking at the particle multiple times for pre-activation. Furthermore, since mineral dust particles are not uniform, the gaps between particle and substrate are very likely much larger than a few nanometers. Having "accidently" a gap where the particle touches the substrate similar to a specific pore size active at that specific supersaturation is unlikely. Pores of a few nanometers, one finds almost only on apparently planar and smooth surfaces but not between a few hundred nanometer to micrometer sized particle touching a smooth substrate. Also, if this would be the case, one would, in principle have always some degree of pre-activation using deposited particles which is not the case. Depositing different mineral types, one measures different ice formation conditions. See e.g. Eastwood et al. (2008), where calcite deposited on a substrate shows vastly different ice formation than Kaolinite. The arguments put forward would also imply that deliquescence and efflorescence data are prone to artifacts as well which hasn't been substantiated. Lastly, even if one argues that there is a gap between particle and substrate in suggested pore size, it is a gap and not a pore and one side of the gap is chemically vastly different compared to the mineral dust particle. The case, that there are pores of specific properties due to having particles deposited is just completely unsubstantiated.

Page 13, line 30: "Edwards. . ..". Please omit and see previous comment.

Page 17, section 5.1: This section should be completely omitted. This is way too speculative to be included. There are so many groups using this technique and an issue like this would have been communicated previously. See comments above.

Page 23, line 3: Statements can be changed in a way: "...indicating the presence of pores. . ." for ". . .suggesting the presence of pores. . .", etc. Again, it is a new concept only. . ..

Page 25, line 17-18: Again, unsubstantiated claims that in all cold stage experiments water is present between particle and substrate causing pre-activation and in principle artifacts. This should be discarded. Bringing this point up over and over in this

manuscript is really detracting.

Technical Corrections:

Page 1, line 14: Maybe omit "severe". Not really a quantitative statement.

Page 1, line 19: Maybe "is" instead of "are".

Page 7, line 3: Maybe "decreases" instead "sinks".

Figure captions 1-3: Captions could be shortened in cases where same data are shown.

References:

Marcolli, C.: Deposition nucleation viewed as homogeneous or immersion freezing in pores and cavities, Atmos. Chem. Phys.,14, 2071–2104, doi:10.5194/acp-14-2071-2014, 2014.

Koop, T., Luo, B. P., Tsias, A., and Peter, T.: Water activity as the determinant for homogeneous ice nucleation in aqueous solutions, Nature, 406, 611–614, doi:10.1038/35020537, 2000.

Koop, T. and Murray, B. J.: A physically constrained classical description of the homogeneous nucleation of ice in water, The Journal of Chemical Physics 145, 211915 (2016); doi: 10.1063/1.4962355

Kanji, Z. A., Florea, O., Abbatt, J. P. D.: Ice formation via deposition nucleation on mineral dust and organics: dependence of onset relative humidity on total particulate surface area, Environ. Res. Lett. 3 (2008) 025004.

Murray, B. J. O'Sullivan, D., Atkinson, J. D., and Webb, M. E.: Ice nucleation by particles immersed in supercooled cloud droplets, Chem. Soc. Rev., 41, 6519–6554, doi:10.1039/c2cs35200a, 2012.

Eastwood, M. L., S. Cremel, C. Gehrke, E. Girard, and A. K. Bertram: Ice nucleation

on mineral dust particles: Onset conditions, nucleation rates and contact angles, J. Geophys. Res., 113, D22203, doi:10.1029/2008JD010639, 2008

---

## Referee Comment (RC2) · Anonymous Referee #2 · 11 Nov 2016

The manuscript by Marcolli reviews previous laboratory experiments on the pre-activation of aerosol particles by the pore condensation and freezing (PCF) mechanism. The PCF mechanism has been introduced by the same author two years ago as a potential ice nucleation pathway for heterogeneous ice formation at temperatures below 235 K and relative humidities below water saturation. Under such conditions, heterogeneous ice formation was before conceptually ascribed to deposition nucleation without involving liquid water. Depending on the temperature for melting and the relative humidity for sublimation, ice trapped in pores from a first nucleation event can facilitate ice crystal growth in a second nucleation event, i.e., lead to pre-activation. The author first describes the conditions for pre-activation in terms of pore size and RH for several pore geometries. Then, previous literature on pre-activation is summarized, and the data are analyzed by categorizing them with respect to the experimental

set-up, the aerosol type, and the pre-activation conditions. Finally, potential scenarios where pre-activation could contribute to ice formation in the atmosphere are outlined.

The manuscript is generally well written and the previous literature thoroughly reviewed. The pre-activation topic, as first emerged around 1950 but then neglected for decades, has received new attention in the past years – and it is a valuable effort and therefore fits well within the scope of ACP to critically review the current state of knowledge in order to stimulate further experimental and theoretical work on this issue. I therefore support the publication in ACP, but have some concerns, as outlined below, which should be addressed before final publication.

Major comments:

1) My first major concern is the widespread use of the term "PCF mechanism" to account for all pre-activation experiments and trajectories discussed throughout the manuscript. If I understand correctly, the PCF mechanism was proposed as kind of challenging hypothesis against the classic view of deposition nucleation at temperatures below 235 K. And there are good reasons for this hypothesis, most importantly the well-documented, strong increase of the ice nucleation efficiency of numerous types of aerosol particles just below 235 K. But the manuscript's title "Pre-activation of aerosol particles by pore condensation and freezing" implies that in all considered examples the PCF mechanism accounts for initial ice formation and pre-activates the particles. In some cases, the pore ice might indeed be formed by the PCF mechanism, but there are numerous examples where there is no need to explicitly invoke this theory. For example, for all wet trajectories discussed in Figs. 1-3, initial ice formation and potential pre-activation occur by droplet activation and immersion freezing somewhere on the particle surface. At least if I understand correctly, ice formation by immersion freezing is not supposed or required to happen directly inside the pore. The susceptibility to pre-activation would then just depend whether ice propagates into the pore or not (e.g. inhibited by a narrow pore opening of an ink-bottle-shaped pore), but is not explicitly caused by the PCF mechanism as defined above. Also for the dry trajectories where

initial ice formation occurs at colder temperatures, there is no need to exclusively infer the PCF mechanism as the formation pathway for pore ice. Wouldn't it be possible that certain pores or void spaces in an aggregate particle can also be filled with ice in a "conventional" deposition nucleation pathway? Instead of writing e.g. on page 9, line 15-16 "... reviewed under the presumption that pre-activation occurred by the PCF mechanism", I would like to see a more general statement that the experiments are analyzed under the assumption that pre-activation is due to the formation and retention of ice in pores, but that there are various mechanisms by which pore ice can be formed, one of them being the newly proposed PCF mechanism.

2) As a second major issue, I would like to see a bit more discussion on the sublimation of ice in pores and whether and for how long ice could survive even in an ice-subsaturated environment. Are there any experimental or modeling studies on that issue? The author argues on the one hand, for example for the dry trajectory shown in Fig. 1, that ice immediately sublimates in the 8 nm-sized cylindrical pore after RHi drops below ice saturation during adiabatic heating. On the other, the generally better pre-activation efficiency observed in the cold stage experiments is always explained by the hypothesis that pore ice is conserved in voids between the substrate and the particle, even if conditioning occurs at RHi values well below 100%. But why should ice located between the particle and the substrate be more stable against sublimation at ice-subsaturated conditions compared to the case where ice is retained in pores within the particle or between particle aggregates? If there is no valid argument for this, such a definite conclusion as e.g. on page 25, line 17-18 cannot be drawn.

Minor comments:

Page 4 & 5 in general: Please also indicate in the main text how the ice and water vapor pressures were calculated, I only discovered this information in the Figure captions.

Page 5, Sect. 3 in general: There is frequent reference to the melting temperature of ice in pores throughout the discussion (computed with Eq. 4). Maybe it would be useful

to include of graph of the pore-diameter-dependent melting temperatures, similar as in Marcolli (2014).

Page 6, lines 15-16: This is one occasion where the immediate sublimation of pore ice at RHi below 100% is assumed (see comment above). But later on (e.g. page 13, line 1 or page 14, line 14,15), it is argued that ice in spaces between a particle and a substrate could trigger ice crystal growth even for RHi « 100% during conditioning.

Page 7, lines 10 – 12: I do not understand the line of argument here.

Page 15, lines 3 – 5: Here, it might be good to refer to the Adler et al. (2013) study, where the formation of porous particles upon freeze-drying was clearly shown with the microscope images.

Page 17, line 15: See above: Why should particularly this ice between particle and substrate survive long exposure to dry conditions and then contribute to pre-activation?

Page 23, line 1 ff: In such a case (when the ash particles are not pre-activated at RHi < 100%), pre-activation would then require a preceding ice nucleation event with the ash particles and not just cooling to low temperatures where pore water could freeze at ice-subsaturated conditions. This would certainly lower the relevance of pre-activation.

Page 24, line 8: Could you include references that meteoritic particles have proven to be poor INPs? I recall e.g. the study by Saunders et al. (2010) where nanoparticles of iron oxide, silicon oxide and magnesium oxide were considered as relatively efficient INPs at T < 220 K.

Saunders, R. W., Möhler, O., Schnaiter, M., Benz, S., Wagner, R., Saathoff, H., Connolly, P. J., Burgess, R., Murray, B. J., Gallagher, M., Wills, R., and Plane, J. M. C.: An aerosol chamber investigation of the heterogeneous ice nucleating potential of refractory nanoparticles, Atmos. Chem. Phys., 10, 1227-1247, doi:10.5194/acp-10-1227-2010, 2010.

Page 26, line 13: I actually like the speculations about the scenarios where preactivation could contribute to atmospheric ice formation in Sect. 6, but a statement in the summary section like "are likely to influence ice cloud formation" is not enough substantiated and should be more clearly denoted as a hypothesis. The same holds for the statement on page 25, line 17-18 as outlined above.

Technical corrections:

Page 1, line 1: aerosol

Page 1, line 19: . . . is perfectly sheltered . . .

Page 3, line 10: humidities

Page 4, line 29: wrong Greek symbol for the density of liquid water

Page 5, line 1: use Greek symbol for the surface tension

Page 6, line 2: sublimating pore ice

Page 6, line 11: liquid water within the pore evaporates

Page 7, line 23/24: ice crystal sublimates

Page 13, line 19: maybe it is meant: among the few samples

Page 26, line 4: relevance . . . depends

Page 43, line 1-2: maybe it is meant: Tcond – conditioning temperature

Page 46, line 4: shouldn't it be 238 K ?

---

## Author Comment (AC1) · 19 Dec 2016

*I thank the reviewer for his/her thoughtful comments, which I address below in italic.*

This manuscript puts forward a literature review-based concept of pre-activation by aerosol particles via the pore condensation and freezing (PCF) mechanism. Application of this mechanism suggests that pre-activation by PCF is constrained by the melting of ice in narrow pores and the sublimation of ice from wide pores. For these reasons, the author argues pre-activation for cylindrical pores is imposed by restrictions on the temperature and relative humidity range. In addition to reviewing previous experimental data sets with regard of finding indications of this concept, the author also puts forward atmospheric scenarios where pre-activation may play a significant role in atmospheric ice formation.

The topic of this manuscript fits within the scope of ACP. The author carefully reviewed the previous literature dealing with pre-activation phenomena. Although, I like the proposed concept and the effort to use previous data for interpretation, I feel some revisions that deal with the general uncertainty of proposed concept and data, are necessary before this manuscript can be published. The author has my full support of publishing this manuscript, hopefully encouraging further experimental investigation of this effect.

As written, the manuscript often reads as if the novel concept is a "fact". One has to keep in mind that there is no experimental in situ proof of the suggested mechanisms for discussed and investigated particles. Considering this, some statements appear "too factual" and thus should be changed in a way to convey the suggestive nature of this discussion.

*This review is intended to have an explorative character by asking the question: assuming that pre-activation is due to ice persisting in pores, what pore properties would be needed to explain the experimental findings? Recent studies carried out with mesoporous silica materials permit to constrain the conditions needed for pre-activation due to pore ice. However, contributions of other mechanisms to the reported cases of pre-activation cannot be excluded. I added a new Section 5.4 (Alternative pre-activation mechanisms) to make clear that the analysis of the experimental data with respect to a mechanism involving pore ice does not preclude other mechanisms leading to pre-activation. In the revision process of the manuscript, I checked the statements for their factuality and emphasized more their suggestive nature.*

For example, the ice formation experiments from the second half of the last century are not well constrained in terms of particle and ice crystal numbers, relative humidity, etc. Often no control or calibration experiments were performed. Considering that even current ice nucleation experiments deviate significantly (see recent data reviews or intercomparison studies), the experimental data can very likely not be used as a definitive support of the proposed concept. This is also indicated by the values in the presented tables which do not include any uncertainties and in many cases the errors, I believe, cannot even be defined or are just very large. Keeping this all in mind, some statements should be more adequately formulated.

*Some of the reviewed papers give detailed description of the experimental conditions including uncertainty ranges others are rather qualitative. Each study should be judged on its own and not be discarded just because of its age.*

For this review, I read Marcolli (2014) that introduces PCF. It is argued that homogeneous freezing occurs in the nanometer-sized pores. From this, as far as I understand, the critical size of the ice embryo fitting inside a pore is derived. However, does homogeneous freezing not also depend on the volume and time? The homogeneous freezing line corresponds to about $J_{hom}=10^{10}$ cm$^{-3}$s$^{-1}$ (Koop et al., 2000). Pores 4 – 20 nm wide and about 16-20 nm deep have a volume of about 1E-19 cm$^3$, resulting in an ice nucleation rate of about $10^{-9}$ s$^{-1}$. Obviously, one would need to wait $10^9$ s at those fixed conditions to observe 1 ice nucleation event in 1 second. The liquid in $10^{18}$ pores would be needed to observe a freezing event in 1 second. Maybe $J_{hom}$ in pores is different but then other aspects/assumptions break down. Very recently Koop and Murray (2016) showed that $J_{hom}$ is not continuously increasing with decreasing temperature, limiting the rate for nucleation to about $10^{12}$ cm$^{-3}$s$^{-1}$. Maybe I am missing here something? My point is that all reported or applied ice nucleation data sets inherently are based on different particle surface areas and experimental time scales and have different pore numbers (and sizes), all of which are mostly unknown or associated with large uncertainties. Thus, it is very unlikely that any of the stated experiments can be used to make a definitive case for pre-activation by PCF.

*Thank you for bringing up this question which I in fact considered when I prepared Figure 3 of Marcolli (2014), but these consideration are not stated in the paper. So I do it here:*

*Figure 3 of Marcolli (2014) shows the freezing data of completely filled pores, most of them determined by differential scanning calorimetry (DSC) with cooling rates of 0.5 - 5 K/min and evaluating the onset of the freezing peak. This experimental data is compared with the freezing temperatures determined from the*

*CNT parameterization by Zobrist et al. (2007) applied to the pore freezing conditions in the DSC and evaluated as a function of pore diameter. For the calculation of the freezing rate, it was assumed that the experiment was carried out at a cooling rate of 0.5 K/min and that the onset of the freezing peak is representative of a frozen pore fraction of 0.01. Since only for the cage-like pores homogeneous ice nucleation is expected (data from Kittaka, 2011; Janssen, 2004; Liu 2007), freezing was assumed to occur in spherical pores. It can be seen from Fig. 3 of Marcolli (2014) that the measured freezing temperatures of homogenous nucleation are in good agreement with the calculated ones. This justifies the assumption that homogeneous ice nucleation is still quite efficient in small volumes as long as the volume is larger than the critical nucleus size given by CNT.*

*With the parameterization by Zobrist et al. (2007) a nucleation rate coefficient of $J_{hom} = 10^{10}$ cm$^{-3}$s$^{-1}$ is reached at a temperature of about 235 K. At 230 K, it is already $J_{hom} = 10^{18}$ cm$^{-3}$s$^{-1}$. These values are much larger than the ones from the new CNT parameterization by Koop and Murray (2016), which was derived using physically consistent parameterizations of the key parameters of CNT, namely, ice-liquid interfacial energy and the diffusion activation energy. However, this parameterization does not seem to be applicable to ice nucleation in pores. In fact, the uncertainties associated with homogeneous ice nucleation rates are notoriously high and even increase the lower the nucleation temperature is. This is due to experimental uncertainties and difficulties. To reach very low freezing temperatures, either the sample has to be cooled very fast or the sample volume has to be very small. To justify their parameterization, Koop and Murray (2016) refer to Laksmono et al. (2015) who applied very high cooling rates to their samples. They found nucleation rate coefficients below $10^{13}$ cm$^{-3}$s$^{-1}$ between 226 - 232 K when they cooled micrometer-sized droplets by 1000 - 10000 K/s. Since cooling rates of $10^7$ K/s lead to vitrification instead of freezing, the maximum nucleation rate coefficient that can be obtained from experiments with microdroplets is $10^{16}$ cm$^{-3}$s$^{-1}$ (Laksmono et al., 2015). This restriction does not apply to nanodrops or nanopores, since here, high nucleation rate coefficients can be reached with lower cooling rates because the volume is smaller. Manka et al. (2012) and Huang and Bartell (1995) reached nucleation rate coefficients of $> 10^{23}$ cm$^{-3}$s$^{-1}$ when they investigated nanometer-sized droplets in the temperature range from 202 K - 228 K. I think, that the relevant experiments to estimate freezing rates in nanopores are the ones that are carried out with small volumes rather than huge cooling rates. The best argument that homogeneous ice nucleation in nanopores indeed occurs is the freezing peak in the DSC experiments with SBA-16 with cage-like pores that are connected by too narrow pores for ice to propagate, so that water has to nucleate in each cage individually. From this it can be concluded that nucleation rates are high enough for pore freezing.*

The same discussion/exercise can be done assuming immersion freezing in a pore by an active site. Immersion freezing and deposition ice nucleation are known to depend on particle surface area (e.g. Kanji et al., 2008). Looking at the literature (e.g. review article by Murray et al. 2012) it looks like "a lot of surface area" has to be provided to detect ice formation. For example, typical experimental particulate surface areas are larger than 1E14 nm2 to observe ice formation. Many pores are needed that contain an active site to be able to reproduce the data sets.

*The probability of immersion freezing in pores has been estimated in Marcolli (2014; pages 2082 – 2083) for ATD using the parameterization by Marcolli et al. (2007). Indeed highly porous particles are needed to render the presence of an active site within a pore probable.*

I am not stating this to cast doubt on the PCF mechanism, which I like and support, but at current stage I recommend to be more careful how to discuss this concept with regard to experimental data. Having said all that, I am not surprised to see some experiments somehow following the presented concept and some not, even if same or similar porous materials were applied. The data sets are just not sufficiently constrained. Statements that a particular approach, such as the cold stage experiment, as discussed in more detail below, is producing potentially erroneous data with respect to pre-activation is, however, unfounded and should be discarded. With present uncertainties and lack of experimental proof, those statements are unjustified. As a matter of fact, these statements detract from the overall nice manuscript.

*Cold stage experiments are not discarded in the discussion of pre-activation. However, it is important to mention possible artifacts, even if further investigations might not substantiate them.*

Page 5-6, section 3: It would be interesting to know how long it takes for ice or water to evaporate from the different pores. This could be done as a function of difference of pore equilibrium RH and ambient RH (and exemplary pore size). This would give an idea if the transient state is important or not. In particular, in an actual cloud with eddies (up/downdraft), the transient state may be a crucial parameter.

*The timescale of pore sublimation is indeed an important question, however, there is no simple answer to it. Recently, some experimental and modeling studies have been published that investigated the rates and processes of pore evaporation. These studies are now summarized in the revised manuscript in the new section 3.2 (Kinetics of pore ice sublimation).*

Page 7, line 10: "However, . . .". This sentence seems to be confusing.

*Thank you for pointing this out. This sentence should read: "However, a cylindrical pore with 4 nm diameter should have a similar ability of pre-activation."*

Page 8, line 25: "A freeze concentrated. . .". How are the water activity values derived?

*The water activity is not derived. It is assumed to be 0.95 when the pore is completely filled as stated in the sentence above.*

> Page 9, line 28: I highly doubt that the freezing point in that type of experiment can be measured to this degree in 1949. This may not be even possible today.

*This is a problem of converting from degree C to Kelvin. To be consistent within the manuscript, all temperatures are given in Kelvin. When the original paper states the temperature in degree C with one decimal place, I transformed to Kelvin by adding 273.15 K. This leads to two decimal places in the converted temperature.*

> Page 10, line 28: "Results of ball milled Iceland spar in the size range from 1 – 15$\mu$m with large numbers from 1 – 3 $\mu$m were presented in most detail: 1 – 5 % of the particles showed pre-activation when kept for 1 min at 84 – 98 % RHi (see Table 3)." This sounds a bit confusing: Did you mean "Results of ball milled Iceland spar particles, in the size range from 1 – 15 $\mu$m with the largest particle numbers in the size from 1 – 3 $\mu$m, were discussed/investigated in most detailine In this case, 1 – 5 % of the particles showed pre-activation when kept for 1 min at 84 – 98 % RHi (see Table 3)."?

*Yes, this was meant. I revised the manuscript accordingly.*

> Page 11, line 22-24: Can it be shown quantitatively that equilibrium was not reached? This is related to my comment above regarding sublimating ice.

*Recent studies indeed indicate that it takes minutes to reach equilibrium. I added the sentence: "Capillary evaporation within such timescales is in accordance with measurements performed with mesoporous silica materials (see Sect. 3.2)."*

> Page 12, line 33: "However, . . .". Please avoid this statement. There is no evidence for this and just speculation. Though the authors of this study did not use microscopic techniques, as far as I recall this work, this is just not a qualified statement. With better experiments in the future, time will tell. One cannot just say a technique is "wrong" when it does not "obey" a new concept.

*I would like to keep the sentence but I will weaken it by stating: "However, as the particle were deposited on glass cover slips, the location of pore ice might also have been voids between the substrate and the particles instead of pores within the particles or between particle aggregates (see also Sect. 5.1)".*

> Page 13, line 18: "Therefore,. . .". Again this is an unsubstantiated statement considering all uncertainties and should be omitted. In fact, Roberts and Hallett observed the particles and ice crystals with a microscope. Some general remarks for this study and following cold stage experiments below:
>
> If ice forms between a particle and substrate, it will move the particle and the sample image would change. Any microscopist would observe and notice this effect and this would have been long established in the community. This is so significant that it would have not been missed. In particular, when looking at the particle multiple times for pre-activation. Furthermore, since mineral dust particles are not uniform, the gaps between particle and substrate are very likely much larger than a few nanometers. Having "accidently" a gap where the particle touches the substrate similar to a specific pore size active at that specific supersaturation is unlikely. Pores of a few nanometers, one finds almost only on apparently planar and smooth surfaces but not between a few hundred nanometer to micrometer sized particle touching a smooth substrate. Also, if this would be the case, one would, in principle have always some degree of pre-activation using deposited particles which is not the case. Depositing different mineral types, one measures different ice formation conditions. See e.g. Eastwood et al. (2008), where calcite deposited on a substrate shows vastly different ice formation than Kaolinite. The arguments put forward would also imply that deliquescence and efflorescence data are prone to artifacts as well which hasn't been substantiated. Lastly, even if one argues that there is a gap between particle and substrate in suggested pore size, it is a gap and not a pore and one side of the gap is chemically vastly different compared to the mineral dust particle. The case, that there are pores of specific properties due to having particles deposited is just completely unsubstantiated.

*To be susceptible to pre-activation, the voids need to be in the low nanometer range. The filling of such narrow pores is not discernable in a light microscope. It will also not lead to a detectable movement of the particle. The study by Eastwood et al. (2008) used a microscope with a 10x objective lens to detect ice crystals growing from mineral dust particles. Figure 3 of this paper shows that with this magnification even a micrometer movement of a particle is not detectable. The study by Eastwood et al. was not designed to detect pre-activation since the samples were only cooled once. This is the case for most studies that investigate deposition nucleation. To detect pre-activation, the same sample must be cooled at least twice without warming above 273 K between cycles.*

*Irregular shaped particles on a substrate may give rise to irregular shaped voids in which water can condense. Such irregularly structured voids might offer suitable geometries for persisting pre-activation.*

*Moreover, they might be swelling. The voids do not need to have specific properties. Just a portion of them needs to be of the right size so that capillary water can condense in them.*

*Particles that deliquesce and effloresce consist of soluble material. Unlike mineral dusts, they dissolve when relative humidity is increased.*

Page 13, line 30: "Edwards*….*". Please omit and see previous comment.

*I omit this sentence here.*

Page 17, section 5.1: This section should be completely omitted. This is way too speculative to be included. There are so many groups using this technique and an issue like this would have been communicated previously. See comments above.

*The difference in the fraction of successfully pre-activated materials observed by cloud chamber experiments compared with cold stage experiments is significant: In the cloud chamber experiments carried out by Fournier d'Albe (1949), Mossop (1956), Day (1958), and Mason and Maybank (1958) 8 – 34% of the tested materials were susceptible to pre-activation. In the cold stage studies by Higuchi and Fukuta (1966), Roberts and Hallett (1968), Edwards and Evans (1971) and Knopf and Koop (2006) 80 – 100 % of the tested materials were susceptible to pre-activation, and pre-activation persisted over long conditioning times at low RH. Capillary water and ice in voids between particles and substrates is an explanation that deserves consideration for this systematic discrepancy between the two techniques. Therefore, I would like to keep this section. It should be a caveat for future studies. The arguments given by the reviewer why such an effect cannot be present are not convincing. I improved in the revised manuscript the explanation why I think that water collected between substrate and particle can be responsible for persistent pre-activation in cold stage experiments.*

Page 23, line 3: Statements can be changed in a way: "...indicating the presence of pores*…*" for "*…*suggesting the presence of pores*…*", etc. Again, it is a new concept only*….*

*I have changed the manuscript accordingly.*

Page 25, line 17-18: Again, unsubstantiated claims that in all cold stage experiments water is present between particle and substrate causing pre-activation and in principle artifacts. This should be discarded. Bringing this point up over and over in this manuscript is really detracting.

*I agree that it is not more than a claim. But I think that this claim should not be discarded. There is indeed a significant difference in the persistence of pre-activation observed in cloud chamber experiments compared with cold stage experiments that deserves a better explanation than to declare old studies untrustworthy. I hope that keeping this statement in the conclusion can stimulate more careful consideration of this issue in future cold stage experiments.*

Technical Corrections:

Page 1, line 14: Maybe omit "severe". Not really a quantitative statement.

*Omitted.*

Page 1, line 19: Maybe "is" instead of "are".

*Corrected.*

Page 7, line 3: Maybe "decreases" instead "sinks".

*Changed.*

Figure captions 1-3: Captions could be shortened in cases where same data are shown.

*I would like to keep the captions the way they are.*

References:

Marcolli, C.: Deposition nucleation viewed as homogeneous or immersion freezing in pores and cavities, Atmos. Chem. Phys.,14, 2071–2104, doi:10.5194/acp-14-2071-2014, 2014.

Koop, T., Luo, B. P., Tsias, A., and Peter, T.: Water activity as the determinant for homogeneous ice nucleation in aqueous solutions, Nature, 406, 611–614, doi:10.1038/35020537, 2000.

Koop, T. and Murray, B. J.: A physically constrained classical description of the homogeneous nucleation of ice in water, The Journal of Chemical Physics 145, 211915 (2016); doi: 10.1063/1.4962355

Kanji, Z. A., Florea, O., Abbatt, J. P. D.: Ice formation via deposition nucleation on mineral dust and organics: dependence of onset relative humidity on total particulate surface area, Environ. Res. Lett. 3 (2008) 025004.

Murray, B. J. O'Sullivan, D., Atkinson, J. D., and Webb, M. E.: Ice nucleation by particles immersed in supercooled cloud droplets, Chem. Soc. Rev., 41, 6519–6554, doi:10.1039/c2cs35200a, 2012.

Eastwood, M. L., S. Cremel, C. Gehrke, E. Girard, and A. K. Bertram: Ice nucleation

on mineral dust particles: Onset conditions, nucleation rates and contact angles, J. Geophys. Res., 113, D22203, doi:10.1029/2008JD010639, 2008

*References:*

*J. Huang and L. S. Bartell, J. Phys. Chem. 99, 3924 (1995).*

*H. Laksmono, T. A. McQueen, J. A. Sellberg, N. D. Loh, C. Huang, D. Schlesinger, R. G. Sierra, C. Y. Hampton, D. Nordlund, M. Beye, A. V. Martin, A. Barty, M. M. Seibert, M. Messerschmidt, G. J. Williams, S. Boutet, K. Amann-Winkel, T. Loerting, L. G. M. Pettersson, M. J. Bogan, and A. Nilsson, J. Phys. Chem. Lett. 6, 2826 (2015).*

*A. Manka, H. Pathak, S. Tanimura, J. Wolk, R. Strey, and B. E. Wyslouzil, Phys. Chem. Chem. Phys. 14, 4505 (2012).*

---

## Author Comment (AC2) · 19 Dec 2016

*I thank the reviewer for his/her thoughtful comments and the corrections, which I address below in italic.*

The manuscript by Marcolli reviews previous laboratory experiments on the pre-activation of aerosol particles by the pore condensation and freezing (PCF) mechanism. The PCF mechanism has been introduced by the same author two years ago as a potential ice nucleation pathway for heterogeneous ice formation at temperatures below 235 K and relative humidities below water saturation. Under such conditions, heterogeneous ice formation was before conceptually ascribed to deposition nucleation without involving liquid water. Depending on the temperature for melting and the relative humidity for sublimation, ice trapped in pores from a first nucleation event can facilitate ice crystal growth in a second nucleation event, i.e., lead to pre-activation. The author first describes the conditions for pre-activation in terms of pore size and RH for several pore geometries. Then, previous literature on pre-activation is summarized, and the data are analyzed by categorizing them with respect to the experimental set-up, the aerosol type, and the pre-activation conditions. Finally, potential scenarios where pre-activation could contribute to ice formation in the atmosphere are outlined.

The manuscript is generally well written and the previous literature thoroughly reviewed. The pre-activation topic, as first emerged around 1950 but then neglected for decades, has received new attention in the past years – and it is a valuable effort and therefore fits well within the scope of ACP to critically review the current state of knowledge in order to stimulate further experimental and theoretical work on this issue. I therefore support the publication in ACP, but have some concerns, as outlined below, which should be addressed before final publication.

Major comments:

1) My first major concern is the widespread use of the term "PCF mechanism" to account for all pre-activation experiments and trajectories discussed throughout the manuscript. If I understand correctly, the PCF mechanism was proposed as kind of challenging hypothesis against the classic view of deposition nucleation at temperatures below 235 K. And there are good reasons for this hypothesis, most importantly the well-documented, strong increase of the ice nucleation efficiency of numerous types of aerosol particles just below 235 K. But the manuscript's title "Pre-activation of aerosol particles by pore condensation and freezing" implies that in all considered examples the PCF mechanism accounts for initial ice formation and pre-activates the particles. In some cases, the pore ice might indeed be formed by the PCF mechanism, but there are numerous examples where there is no need to explicitly invoke this theory. For example, for all wet trajectories discussed in Figs. 1-3, initial ice formation and potential pre-activation occur by droplet activation and immersion freezing somewhere on the particle surface. At least if I understand correctly, ice formation by immersion freezing is not supposed or required to happen directly inside the pore. The susceptibility to pre-activation would then just depend whether ice propagates into the pore or not (e.g. inhibited by a narrow pore opening of an ink-bottle-shaped pore), but is not explicitly caused by the PCF mechanism as defined above. Also for the dry trajectories where initial ice formation occurs at colder temperatures, there is no need to exclusively infer the PCF mechanism as the formation pathway for pore ice. Wouldn't it be possible that certain pores or void spaces in an aggregate particle can also be filled with ice in a "conventional" deposition nucleation pathway? Instead of writing e.g. on page 9, line 15-16 "... reviewed under the presumption that pre-activation occurred by the PCF mechanism", I would like to see a more general statement that the experiments are analyzed under the assumption that pre-activation is due to the formation and retention of ice in pores, but that there are various mechanisms by which pore ice can be formed, one of them being the newly proposed PCF mechanism.

*I agree with the reviewer. I changed the text in the manuscript accordingly.*

*The revised manuscript carries now the title: "Pre-activation of aerosol particles by ice preserved in pores"*

2) As a second major issue, I would like to see a bit more discussion on the sublimation of ice in pores and whether and for how long ice could survive even in an ice-subsaturated environment. Are there any experimental or modeling studies on that issue? The author argues on the one hand, for example for the dry trajectory shown in Fig. 1, that ice immediately sublimates in the 8 nm-sized cylindrical pore after RHi drops below ice saturation during adiabatic heating. On the other, the generally better pre-activation efficiency observed in the cold stage experiments is always explained by the hypothesis that pore ice is conserved in voids between the substrate and the particle, even if conditioning occurs at RHi values well below 100%. But why should ice located between the particle and the substrate be more stable against sublimation at ice-subsaturated conditions compared to the case where ice is retained in pores within the particle or between particle aggregates? If there is no valid argument for this, such

a definite conclusion as e.g. on page 25, line 17-18 cannot be drawn.

*The timescale is indeed an important issue. However, there is no simple way to calculate or estimate the sublimation rate of ice in pores. There are some recent papers that treat this question. Based on these studies, I added a new section 3.2 to the revised manuscript with the title "Kinetics of pore ice sublimation".*

Minor comments:

Page 4 & 5 in general: Please also indicate in the main text how the ice and water vapor pressures were calculated, I only discovered this information in the Figure captions.

*I added the following sentence to the first paragraph of section 3.1: "Ice saturation and water saturation are calculated with the parameterizations from Murphy and Koop (2005)."*

Page 5, Sect. 3 in general: There is frequent reference to the melting temperature of ice in pores throughout the discussion (computed with Eq. 4). Maybe it would be useful to include of graph of the pore-diameter-dependent melting temperatures, similar as in Marcolli (2014).

*Instead of reproducing the Figure of Marcolli (2014), I prefer to explicitly refer to it at the end of Section 2. Moreover, the pore melting temperatures are indicated in Figures 1 – 3 of this review by the high temperature end of the dashed portion. Water in pores of 2 nm and 1 nm remains liquid. This is now explicitly stated in the figure captions of Figs. 1 – 3.*

Page 6, lines 15-16: This is one occasion where the immediate sublimation of pore ice at RHi below 100% is assumed (see comment above). But later on (e.g. page 13, line 1 or page 14, line 14,15), it is argued that ice in spaces between a particle and a substrate could trigger ice crystal growth even for RHi « 100% during conditioning.

*For the discussion of the trajectories, thermodynamic equilibrium was assumed. This is stated at the beginning of Sect. 3.1. To make this clearer, the statement on page 6, lines 15 – 16 is changed to: "Therefore, in a cylindrical pore of 8 nm, no persistent pre-activation occurs for T < 233 K because of the sublimation of the pore ice."*

*When irregularly formed particles are deposited on substrates in cold stages, voids with narrow opening may form between the substrate and the deposited particles. These voids can swell when they fill with liquid water and should be able to keep ice below ice saturation analogously to the case of swelling pores discussed in Fig. 3. Thus, a thermodynamically stable state is assumed and the pore ice should be preserved permanently.*

Page 7, lines 10 – 12: I do not understand the line of argument here.

*This sentence is improved in the revised manuscript. It now reads: "However, a cylindrical pore with 4 nm diameter should have a similar ability of pre-activation".*

Page 15, lines 3 – 5: Here, it might be good to refer to the Adler et al. (2013) study, where the formation of porous particles upon freeze-drying was clearly shown with the microscope images.

*I added the following sentence to the revised manuscript: "This hypothesis was confirmed by Adler et al. (2013) who showed that freeze drying leads to highly porous particles."*

Page 17, line 15: See above: Why should particularly this ice between particle and substrate survive long exposure to dry conditions and then contribute to pre-activation?

*I explain this hypothesis better by reformulating: "When irregularly formed particles are deposited on substrates in cold stages, voids with narrow openings may form between the substrate and the deposited particles. These voids are likely to swell when they fill with liquid water and should be able to keep ice below ice saturation analogously to the case of swelling pores discussed in Fig. 3."*

Page 23, line 1 ff: In such a case (when the ash particles are not pre-activated at RHi < 100%), pre-activation would then require a preceding ice nucleation event with the ash particles and not just cooling to low temperatures where pore water could freeze at ice-subsaturated conditions. This would certainly lower the relevance of pre-activation.

*I agree. The ash of the* Eyjafjallajökull *eruption needs to have gone through a cirrus cloud to become pre-activated. Indeed, cirrus clouds were present at that time (Schumann et al., 2010). However, the porosity of volcanic ashes is variable. In other cases, pre-activation might also be possible at ice-subsaturated conditions.*

Page 24, line 8: Could you include references that meteoritic particles have proven to be poor INPs? I recall e.g. the study by Saunders et al. (2010) where nanoparticles of iron oxide, silicon oxide and magnesium oxide were considered as relatively efficient INPs at T < 220 K.

Saunders, R. W., Möhler, O., Schnaiter, M., Benz, S., Wagner, R., Saathoff, H., Connolly, P. J., Burgess, R., Murray, B. J., Gallagher, M., Wills, R., and Plane, J. M. C.: An aerosol chamber investigation of the heterogeneous ice nucleating potential of refractory nanoparticles, Atmos. Chem. Phys., 10, 1227-1247, doi:10.5194/acp-10-1227-2010, 2010.

*Biermann et al. (1996), and Mason and Maybank (1958) investigated the ice-nucleating ability of meteoritic material. I added these citations to the revised manuscript.*

Page 26, line 13: I actually like the speculations about the scenarios where pre-activation could contribute to atmospheric ice formation in Sect. 6, but a statement in the summary section like "are likely to influence ice cloud formation" is not enough substantiated and should be more clearly denoted as a hypothesis. The same holds for the statement on page 25, line 17-18 as outlined above.

*I agree that these are speculations and I weaken the statement in the summary and conclusions section as requested by the reviewer.*

Technical corrections:

*Thanks for the corrections*

Page 1, line 1: aerosol     *corrected*

Page 1, line 19: . . . is perfectly sheltered . . . *corrected*

Page 3, line 10: humidities     *corrected*

Page 4, line 29: wrong Greek symbol for the density of liquid water  *corrected*

Page 5, line 1: use Greek symbol for the surface tension     *corrected*

Page 6, line 2: sublimating pore ice     *corrected*

Page 6, line 11: liquid water within the pore evaporates  *corrected*

Page 7, line 23/24: ice crystal sublimates    *corrected*

Page 13, line 19: maybe it is meant: among the few samples  *yes, corrected*

Page 26, line 4: relevance . . . depends *corrected*

Page 43, line 1-2: maybe it is meant: Tcond – conditioning temperature *yes, corrected*

Page 46, line 4: shouldn't it be 238 K ?  *yes, corrected*

---

## Author Response (AR2)

*I would like to thank the referees to review the manuscript again. Below are my responses in italic*

**Reviewer #1:**

My suggestions/corrections have been addressed satisfactorily, there remain only a few, mostly technical corrections that the author should take into account.

1) Page 6, line 8 (in the revised manuscript version with applied changes): after the statement "Transient persistence of sublimating pore ice is neglected" the author could add a reference to the new section 3.2 where the kinetics of pore ice sublimation are elaborated.

*I did this.*

2) Page 18, line 29: In line with the more careful interpretation of potential ice between substrate and particle, the author should state here: "These voids could swell when they fill with liquid water and be able to keep ice below ice saturation ..." (instead of "are likely to swell")

*I changed the sentence according to the reviewer's suggestion.*

3) Page 20, line 22: Please assign the Kaufmann et al. (2016) reference to either (a) or (b).

*Thanks for pointing this out. It is (a).*

4) Two are two places where still the expression "pre-activation due to the PCF mechanism" instead of the new "pre-activation due to ice preserved in pores" is used, these are on page 22, line 20 and on page 27, line 17.

*Thanks for pointing this out. I revised the formulations.*

5) The heading of the new section 5.4 does not appear in the pdf-version with the applied changes.

*The line break was missing as pointed out by reviewer #2.*

5   **Reviewer #2:**

This is the evaluation of the revised manuscript by C. Marcolli entitled "Pre-activation of aerosol particles by ice preserved in pores". I appreciate the rebuttal of my previous criticism by the author. The manuscript has been improved and can be published. However, I still partly disagree with the author on the emphasis that the cold stage experiments are prone to artifacts while giving the impression the other techniques are not. I can imagine that, e.g., optical microscopy cannot tell about potential liquid water or ice between particles and substrate (on small scales). However, looking at recent articles including electron microscopy, the ice crystals observed are usually rather large (10-100s of micrometer). This size is larger than the size of the aerosol particles on the substrate and thus would alter the visual appearance of the sample which should not go unnoticed (I suppose, in particular when repeating the experiment with same sample). I am not necessarily disagreeing with the author but the discussion feels not well-balanced since the other techniques' caveats are not much mentioned.

*I alleviated the emphasis that the cold stage experiments are prone to artifacts even more by following the suggestion (2) of reviewer #1. I still would like to keep this section as a caveat for future work rather than as a fact.*

Technical corrections:
p. 22. Line 30: line break missing for section 5.4

*Thank you for pointing this out. I added the line break.*

[revised manuscript text omitted]